# Human single neuron activity precedes emergence of conscious perception

Hagar Gelbard-Sagiv [1,2,3,4], Liad Mudrik [1,4,5], Michael R. Hill[1,2], Christof Koch[1,6] & Itzhak Fried[2,7]

Identifying the neuronal basis of spontaneous changes in conscious experience in the absence of changes in the external environment is a major challenge. Binocular rivalry, in which two stationary monocular images lead to continuously changing perception, provides a unique opportunity to address this issue. We studied the activity of human single neurons in the medial temporal and frontal lobes while patients were engaged in binocular rivalry. Here we report that internal changes in the content of perception are signaled by very early (~-2000 ms) nonselective medial frontal activity, followed by selective activity of medial temporal lobe neurons that precedes the perceptual change by ~1000 ms. Such early activations are not found for externally driven perceptual changes. These results suggest that a medial fronto-temporal network may be involved in the preconscious internal generation of perceptual transitions.

[1] Division of Biology, California Institute of Technology, Pasadena, 91126 CA, USA. [2] Department of Neurosurgery, David Geffen School of Medicine and Semel Institute for Neuroscience and Human Behavior, University of California, Los Angeles, 90095 CA, USA. [3] Department of Physiology and Pharmacology, Sackler School of Medicine, Tel Aviv University, Tel Aviv 6997801, Israel. [4] Sagol School of Neuroscience, Tel Aviv University, Tel Aviv 6997801, Israel. [5] School of Psychological Sciences, Tel Aviv University, Tel Aviv 6997801, Israel. [6] Allen Institute for Brain Science, Seattle, WA 98109, USA. [7] Functional Neurosurgery Unit, Tel-Aviv Medical Center and Sackler School of Medicine, Tel-Aviv University, Tel-Aviv 6423906, Israel. Correspondence and requests for materials should be addressed to H.G-S. (email: hagar.sagiv@gmail.com)

One of the greatest challenges of cognitive neuroscience is bridging the gap between the binary activity of single neurons and the complexity and vividness of conscious experience. To date, only a few studies have addressed this question in humans, using single neuron recordings from the medial temporal lobe (MTL)[1–3]. In these studies, conscious perception was manipulated using three paradigms: flash suppression, in which the perception of an image presented to one eye is suppressed by flashing a different image to the other eye[4]; backward masking, in which a briefly presented image is suppressed by the immediate presentation of a mask image[5]; and the attentional blink, in which the second of two target stimuli appearing in close succession is often not perceived[6]. Firing patterns of human MTL neurons in response to these paradigms correlated with conscious perception, responding stronger and earlier when the patient perceived the stimulus compared to when the stimulus was perceptually suppressed[1–3]. Notably, the response of MTL neurons started 200–300 ms after the external manipulation that led to the change in perception, raising the question of whether these neurons have a role in generating the percept. However, in these studies, the perceptual content was externally driven by the experimental manipulation, rather than by internal processes. Thus, the involvement of MTL neurons in the internal generation of conscious perception could not be assessed using these paradigms.

As opposed to the above paradigms, in binocular rivalry[7,8] (BR), perception alternates while the stimulus is constant and stationary. Due to the absence of external changes, BR provides an exceptional opportunity to detect internally driven changes in perception. In BR, a different image is presented to each eye, inducing involuntary stochastic perceptual alternations between the two associated percepts, with periods of exclusive dominance of each image and transition periods of mixed percept (piecemeal). In a series of seminal studies[9–11], Logothetis and colleagues used BR while tracking the firing rate (FR) of neurons from primary visual areas up to inferior temporal cortex in monkeys. They reported more cells correlating with the monkeys' percept when ascending the visual hierarchy, from 20% of cells in primary visual areas to 90% of the cells in inferior temporal cortex[12]. But what happens further in the processing chain, in MTL neurons? Given the previously demonstrated role of these neurons in conceptual representations[13,14], are they causally involved in the perceptual alternations observed in BR?

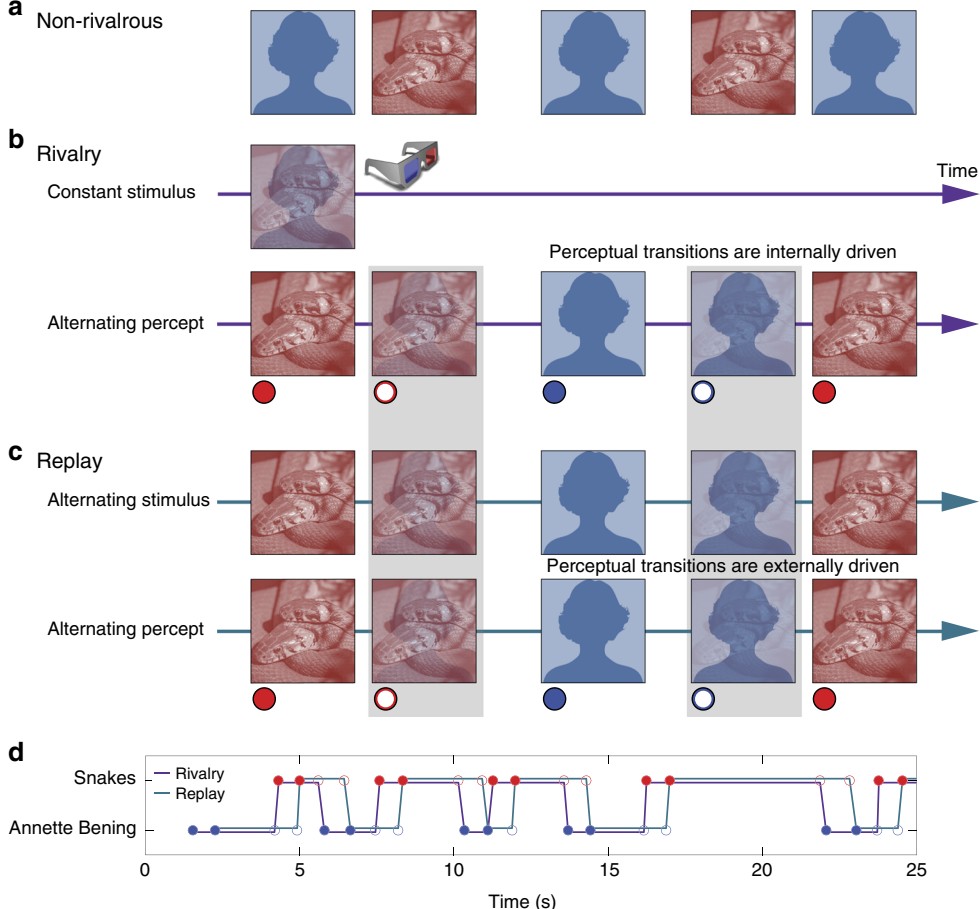

**Fig. 1** Experimental paradigm and behavior. **a** Non-rivalrous condition. Images were presented binocularly to both eyes, hence no binocular conflict. In this example a picture of snakes and a picture of the actress Annette Bening (marked by a placeholder due to copyrights issues) were used. **b** Binocular rivalry condition. A distinct constant image presented to each eye using red-blue goggles (top), resulted in spontaneous perceptual alternations (bottom) reported by the patients by holding (filled circles) and releasing (empty circles) two assigned buttons, for the beginning of the dominance and transition periods, respectively. Gray rectangles mark the report of transition onset towards the suppressed image (end of dominance period of the other image; button release), which is the behavioral event on which the analysis was focused. **c** Matched-duration replay condition. Internally driven perceptual alternations during rivalry were mimicked by gradual and duration-matched physical (externally driven) alternations on screen, presented binocularly to both eyes, with an identical behavioral task. **d** Rivalry (purple) and replay (cyan) behavioral report example. Note how closely the replay reports follow the rivalry reports with a delay of 0.7 ± 0.1 s. An amygdala neuron from this session is presented in Fig. 3

A similar question relates to the role of different frontal areas in internally driven perceptual alternations, which is highly debated[15–24]. This question is of special interest, given the more general controversy among leading theories of consciousness[25–30] regarding the role of frontal areas in generating conscious perception.

Here we recorded from human medial-temporal and medial-frontal lobe neurons in neurosurgical patients, implanted with intracranial electrodes for clinical purposes. Patients were engaged in either BR, where perceptual alternations were internally driven, or in a matched-duration replay condition, in which the perceptual alternations were externally driven by an actual stimulus change. This opportunity to measure single cell activity in humans able to readily report their subjective experience allowed us to track the neural events that precede spontaneous alternations in perception. We show that neurons in the pre-supplementary motor and anterior cingulate areas are active almost 2 s prior to the reported emergence of conscious percepts during BR. This early activation is followed by selective activity of MTL neurons that precedes the perceptual alternations by 1 s. Such early activity is not found when perceptual alternations are externally driven by an actual stimulus change. Thus, medial frontal and medial temporal activity are persumably part of the chain of events that leads to an internal perceptual transition during rivalry.

## Results

**Experimental design**. Nine pharmacologically intractable epilepsy patients implanted with intracranial depth electrodes to localize the focus of seizure onset for potential surgical cure participated in 20 sessions of the experiment. For each patient, placement of the depth electrodes was determined exclusively by clinical criteria[31].

Patients first participated in a selectivity screening session, where they were presented with a large number of images to find ones that elicit selective responses in MTL neurons[32]. Images that induced the strongest responses were paired with images that did not induce responses. These image pairs were used in the following BR session, which took place a few hours later.

Each block of the BR session began with 3–24 ($M = 6.45 \pm 1.71$) non-rivalrous presentations of each of the two images to both eyes (non-rivalrous condition; Fig. 1a) in a pseudo-random order. Patients pressed two pre-assigned buttons in response to the two images. After they reached eight successive correct responses to each image, the non-rivalrous condition was terminated. This non-rivalrous condition was aimed at training patients to accurately report the content of their percept, as well as to reassess the selectivity of the targeted units. During the rivalry condition, the two images were presented monocularly, one to each eye. Patients reported four perceptual events: dominance onset of each of the two images, defined as the time at which this image started to be exclusively perceived; and transition onset for each of the images, defined as the instant at which that image started to emerge into awareness (i.e. become visible) after being perceptually suppressed. Patients were instructed to report transition onset as soon as something in the dominant image started to change and not to wait for seeing a clear part of the emerging image—any change in the dominant percept should have been reported as transition onset. Importantly, our analysis focused on these transition onset events, marking the emergence of a new percept. Reports of dominance and transition onsets were done by holding and releasing the two assigned buttons, respectively[11] (Fig. 1b). As opposed to the non-rivalrous condition, here the patients were asked to report their perceptual changes as quickly as possible (Methods).

These reports were then used to create a matched-duration replay condition in which the same stimulus was presented to both eyes in an order specified by the reports during rivalry (Fig. 1c): during reported dominance periods, the relevant image was presented, and during transition periods (times between transition onset to dominance onset), the transparency of the previous image was linearly ramped up to 100% while the transparency of the next image was linearly ramped down to zero. As opposed to many previous BR studies that used instantaneous replay very different from actual rivalry[20], this stimulation was designed to produce a perception that closely mimics rivalry, resulting in a matched sequence of motor responses in the two conditions (see for example, Fig. 1d). Note that patients were not informed about the difference between the conditions, and performed the same behavioral task. Patients were also not debriefed about this at the end of the experiment, in order to keep them naïve for additional sessions.

**Rivalry and replay time courses**. Rivalry blocks lasted either 90 or 120 s (114.50 ± 12.76 s mean ± standard deviation; SD). Overall, patients completed rivalry and replay blocks with 1–4 ($M = 2.70 ± 0.73$) different pairs of images per session, including 2−5 ($M = 2.67 ± 0.66$) rivalry and 1–2 ($M = 1.08 ± 0.24$) replay blocks for each image pair. On average, subjects had 44.87 ± 23.29 alternations per image pair during rivalry, with each dominance period lasting 3.59 ± 1.50 s. Twenty-four percent of the alternations were incomplete (i.e., did not lead to full dominance of the emerging image, but rather to the return of the previously dominant one). Averaged transition duration was 1.68 ± 1.32 s. Patients' predominance score was calculated by dividing the total dominance time of one of the images by the total dominance time for both images. On average, patients' percepts were equally distributed between the two images (predominance score = 0.50 ± 0.05; $t(53) = 0.40$; $p = 0.69$; 95% CI = [0.45 0.53]; two-tailed $t$ test against 0.5). Note that images were switched between the eyes in the middle of the rivalry condition to minimize ocular dominance influence on image dominance distribution (Methods). Frequency histograms of the relative dominance durations—i.e., normalized by dividing by the average dominance duration for each image—fit a gamma distribution[33] (Supplementary Fig. 1).

For each pair of images, a matched-duration replay block was generated, based on the behavioral responses during one of the rivalry blocks (Methods). Mean dominance duration in the replay condition was 4.91 s (SD = 2.23 s; compared to 3.43 s, SD = 1.27 s in the corresponding rivalry block; $W(7) = 34$, $z = 2.24$, $p = 0.025$; Wilcoxon signed-rank two-tailed test) and transition duration was 1.12 s (SD = 0.87 s; compared to 1.49 s, SD = 1.08 s in the corresponding rivalry block; $W(7) = 36$, $z = 2.52$, $p = 0.008$; Wilcoxon signed-rank two-tailed test). The shorter transition times during replay compared to rivalry probably stem from the time it takes the change in transparency to become noticeable on both ends of the transition period. Importantly, this delay should make it easier to find early neuronal responses in the replay condition with respect to the transition report (button release), as during a replayed transition – but not during internally generated transition – the neurons can respond to the physical changes that precede the perceptual change.

**MTL activity precedes perceptual transitions**. We recorded from a total of 402 MTL (166 single[32]) units (see Supplementary Table 1 and Supplementary Fig. 2). 75 (19%) MTL units responded selectively to at least one of the images in at least one of the image pairs during the non-rivalrous condition (120 selective responses, 15 of which were for both images of the pair; see Methods). This relatively low fraction of responsive units

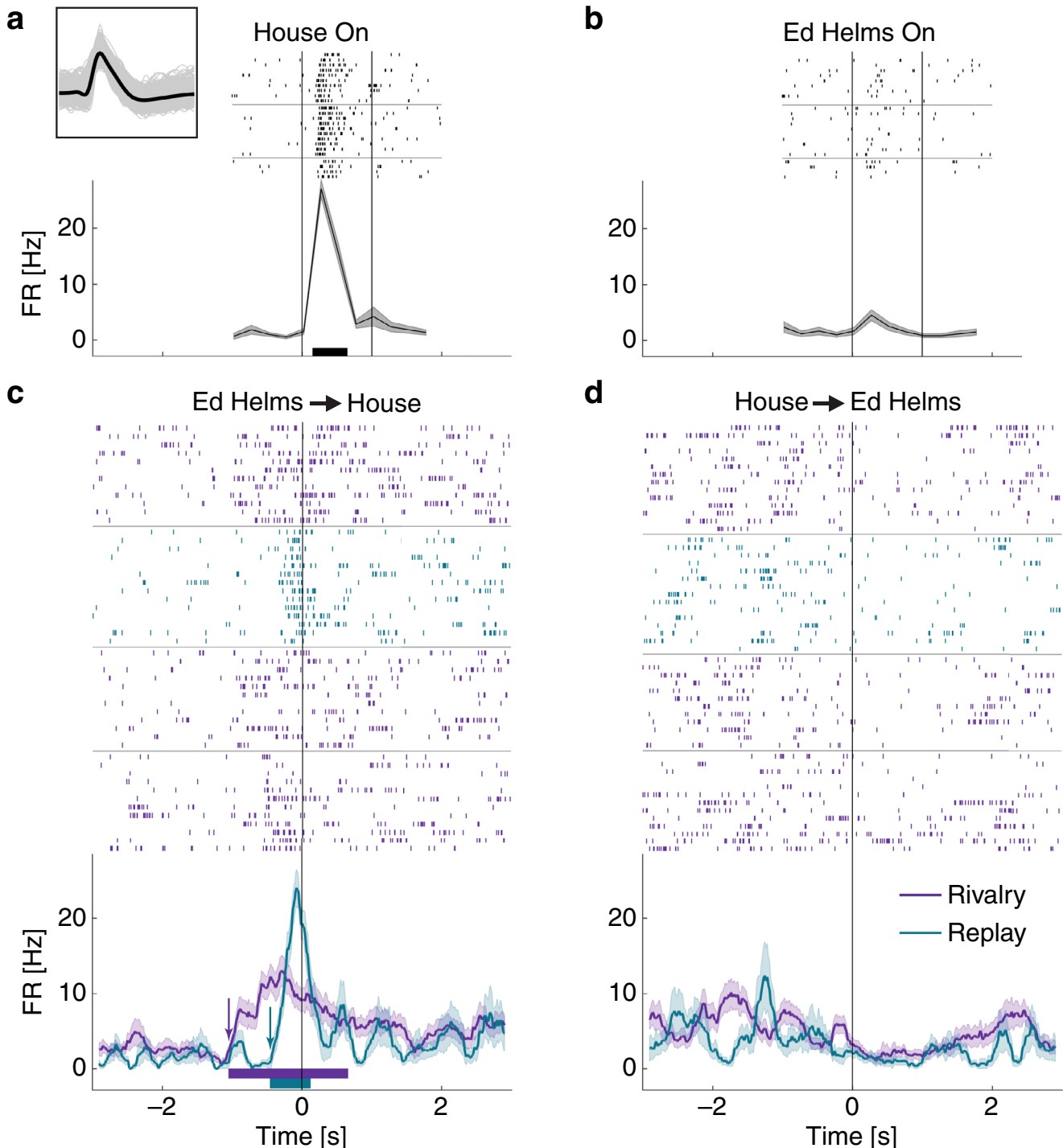

**Fig. 2** Activity of a single-unit in parahippocampal gyrus during rivalry and replay. **a**, **b** Responses to the non-rivalrous presentation of a house (**a**) or the actor Ed Helms (**b**) images. Raster plots (order of trials is from top to bottom) and post-stimulus time histogram (PSTH; 250 ms bins) are given for each image; vertical lines indicate image onset and offset; horizontal lines separate different blocks; black horizontal bars denote significant activity relative to baseline. Inset presents waveforms of this unit. **c** Neuronal firing around the report of transition onset to the house image ($t = 0$; end of Ed Helms image exclusive dominance) during rivalry (purple) and replay (cyan). PSTHs are computed with a moving square-window of 200 ms. Shaded areas represent standard error (SE). Purple and cyan horizontal bars denote periods where the instantaneous FR was significantly different than baseline in rivalry and replay, respectively (onset of these periods marked by arrows). Note the earlier activation during rivalry as compared to replay. **d** Neuronal firing around the report of transition onset to the Ed helms image

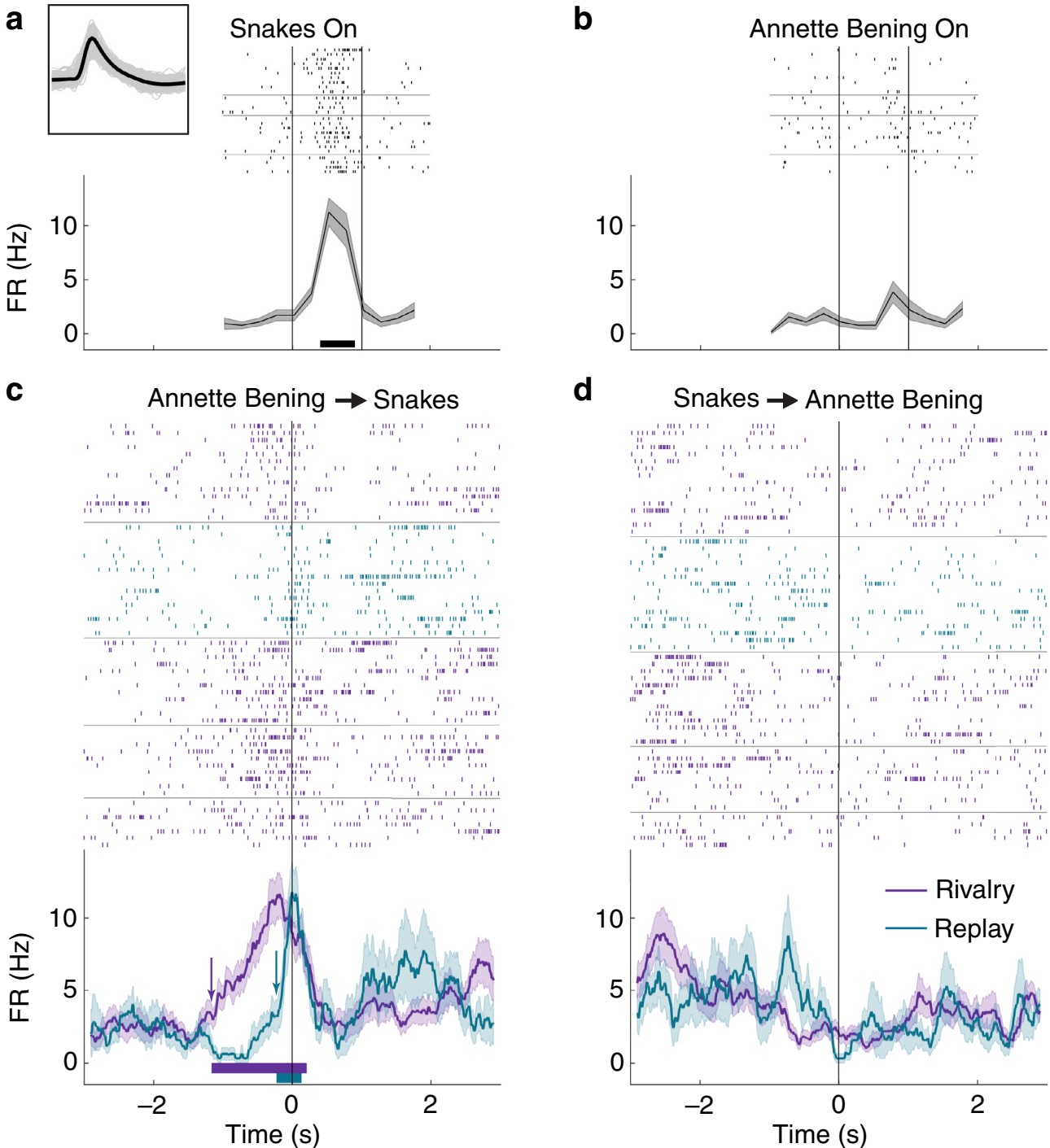

**Fig. 3** Activity of a single-unit in amygdala during rivalry and replay. Conventions as in Fig. 2. **a**, **b** Responses to the non-rivalrous presentation of snakes (**a**) or the actress Annette Bening (**b**) images. **c** Neuronal firing around the report of transition onset to the snakes image (*t* = 0; end of Annette Bening image exclusive dominance) during rivalry (purple) and replay (cyan). **d** Neuronal firing around the report of transition onset to the Annette Bening image. Note the strikingly different onset of firing in rivalry and replay. Behavioral reports from this session are presented in Fig. 1d

is typical of pre-screening based studies[2,3,32], as units that responded in the pre-screening session might not be recorded anymore, or change selectivity in the main experimental session that takes place a few hours later. This low yield is all the more expected here given the long duration of each rivalry and replay block and the limited time with the patients, which allowed the use of only 4–6 images per session on average. Responses were either positive (i.e. increase in FR above baseline; 86 responses) or

negative (decrease below baseline FR; 34 responses); Negative responses were also previously reported in the human MTL[34,35] and medial frontal cortex[36]. For each pair of images, the image with the stronger response during the non-rivalrous condition was defined as the unit's "preferred image".

Out of these responsive units, we identified units that significantly increased or decreased their FR around the onset of the transition to the unit's preferred image using permutation

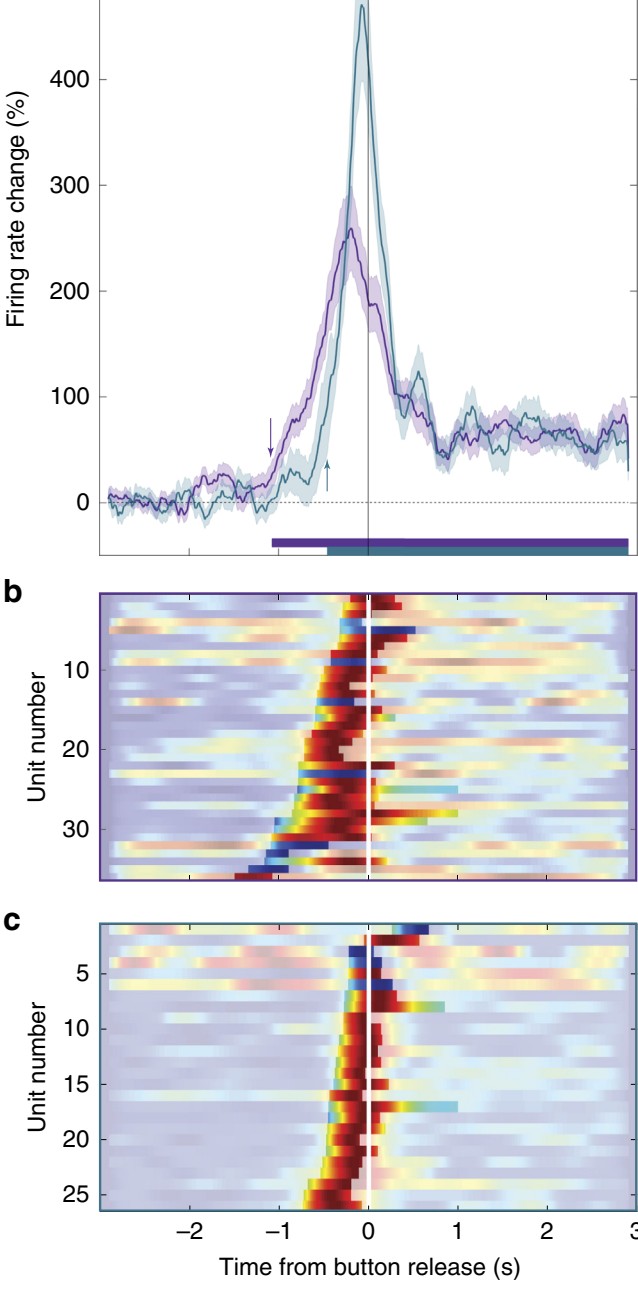

**Fig. 4** MTL population-level activity started earlier during rivalry than during replay. **a** FR percent change (mean ± SE) around the report of perceptual transition onset to the preferred image is averaged across all MTL rivalry-active (purple; $n = 36$) and replay-active traces (cyan; $n = 26$). Traces with significant decrease below baseline FR ($n = 8$ rivalry-active and $n = 4$ replay-active traces; see bluish highlighted periods in (**b**) and (**c**)) were inverted to enable averaging. Thick horizontal bars denote times where population activity significantly diverged from baseline (onset of these periods marked by arrows); note the ~600 ms difference in onset between rivalry and replay. The normalized FR time courses used to create the average responses are presented in color code (dark blue = 0; dark red = 1) for all MTL rivalry-active (**b**) and replay-active (**c**) units, with periods of significance at the unit level highlighted

statistics (see Methods). Note that all transition onsets to the unit's preferred image, whether complete or incomplete, were taken into account; as long as an image starts to emerge into consciousness after being perceptually suppressed (marked by transition onset report), it should be accompanied by a neural event, whether that image gains full dominance or not. The permutation analysis allowed us to both identify active units, and time their activity relative to the behavioral report. Thirty-one MTL units were active during rivalry and/or replay (Supplementary Table 1).

For example, Fig. 2 depicts a parahippocampal unit, which responded selectively to a picture of a house (Fig. 2a) but not to a picture of the actor Ed Helms (Fig. 2b) in the non-rivalrous condition. During rivalry, this unit started to fire about a second prior to the report of the onset of perceptual transition to the house image (Fig. 2c, purple). On the contrary, when the subject reported transitioning to the Ed Helms image (Fig. 2d, purple), the unit's FR was substantially reduced. Interestingly, in the externally driven replay condition, the activity preceding the perceptual switch started almost 600 ms later than during rivalry (Fig. 2c, cyan).

This anticipatory activity during rivalry started even earlier in an amygdala unit recorded from another patient, selective to a picture of snakes (Fig. 3a) and not to a picture of the actress Annette Bening (Fig. 3b). Here, the FR increased more than a second before the report of the perceptual transition to the snakes image but not before the transition to the Annette Bening image (compare purple traces on Fig. 3c, d). During replay, on the other hand, the unit started to fire much later, only at ~150 ms before the report (Fig. 3c, cyan). The same pattern of earlier anticipatory activity in rivalry compared to replay was found regardless of the type of response, that is, also for units that responded by decreasing their FR in the non-rivalrous condition (e.g. Supplementary Fig. 3 from another unit in the same patient).

The above analysis reveals the times at which the activity of each unit started to diverge significantly from baseline. For MTL units, this happened earlier with respect to the reported transition onset to the unit's preferred image during rivalry ($-604 \pm 317$ ms; $n = 36$ traces; Figure 4b, significance periods highlighted) than during replay ($-269 \pm 207$ ms; $n = 26$ traces; Fig. 4c; $t(60) = 4.70$, $p < 0.0001$, 95% CI = [$-478$ $-192$ms], two-tailed independent samples $t$ test). Note that seven out of the 31 recorded units were active in more than one image pair in rivalry and/or replay. As the activity patterns were often different for the different image pairs, we did not collapse across pairs, but rather considered each activity "trace" (i.e., a certain unit responding to a certain image pair) as an independent sample. This led to 36 rivalry-active traces and 26 replay-active traces. To further assess the strength of this difference and the sufficiency of the evidence, we computed the Bayes factor (BF) for this effect (Methods), testing the likelihood of a difference between rivalry and replay activity onsets (H1) vs. the lack of such difference (H0). A BF of 1123 was found, suggesting extreme evidence for the existence of a difference between rivalry and replay activity onsets[37].

In addition, we used an automatic response detection analysis[3,38] (Poisson spike train analysis; see Methods), to detect activity onset on a trial-by-trial basis (see Supplementary Fig. 4a for an example of the application of this method on the data from Fig. 2). Note that this analysis cannot detect decreases in FR; thus, only units that showed increase in FR in either rivalry or replay were included in this analysis ($n = 33$ traces). Accordingly, we used this analysis as an additional way to examine the data and assess the robustness of the main analysis above. Though this analysis is not ideal for this type of continuous data, that does not contain a clear baseline period essential for this analysis, it yielded similar results: an earlier activity onset during rivalry ($-526 \pm$

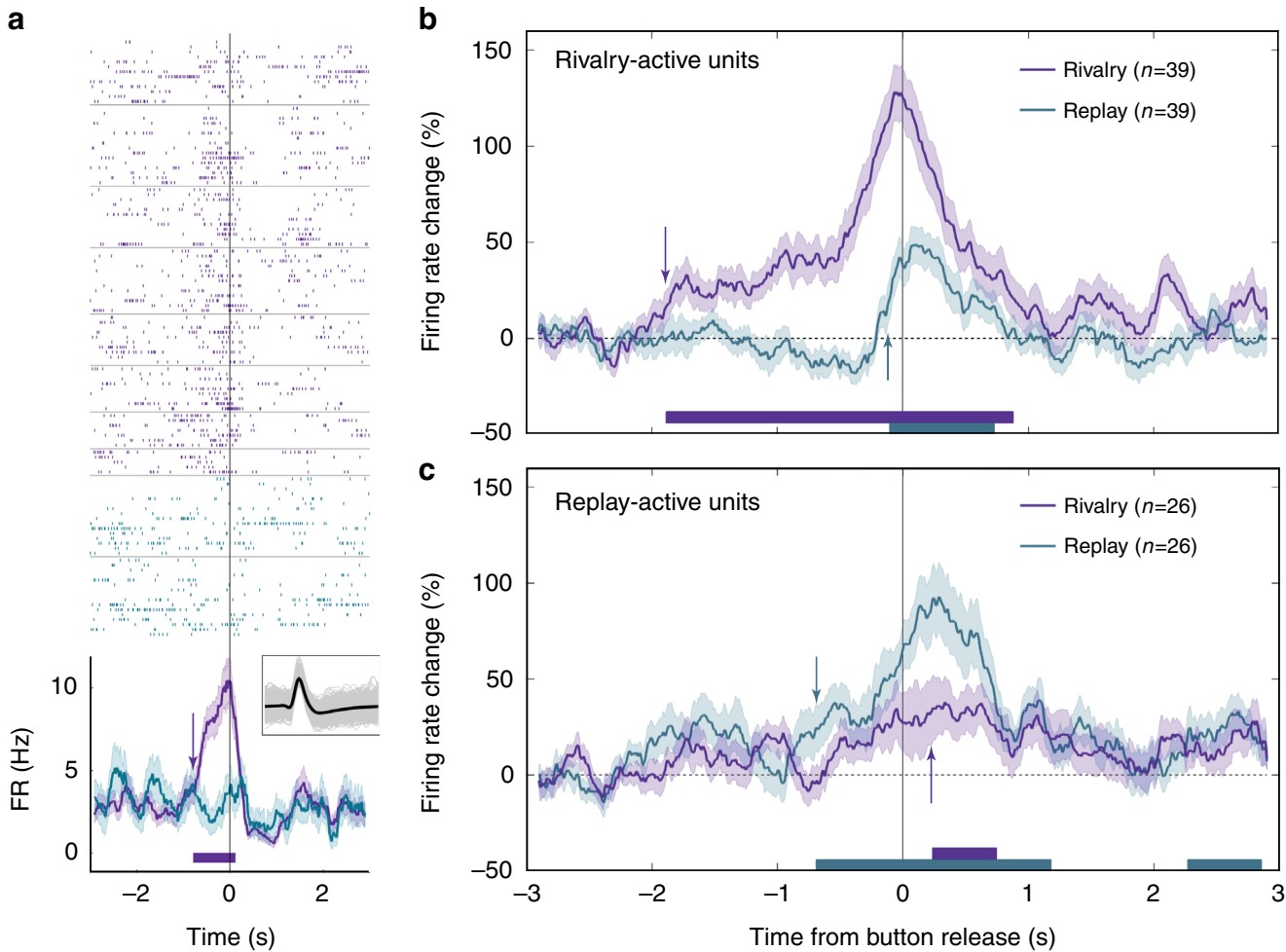

**Fig. 5** ACC and preSMA activity during rivalry and replay. **a** Firing of an ACC multi-unit around the report of a perceptual transition onset during rivalry (purple) and replay (cyan); convention as in Fig. 2c. Note that this anticipatory activity was nonselective and accompanied the transition to any image. Average FR percent change (mean ± SE) across all rivalry-active units (**b**) and replay-active units (**c**); conventions as in Fig. 4a. Note the very early anticipatory activity (almost 2 s before the report) of the rivalry-active units during rivalry as compared to replay

214 ms; median ± SD) as compared to replay (−278 ± 300 ms; $t$(23) = 4.39, $p$ = 0.0002, 95% CI = [−341 −123ms], two-tailed paired $t$-test; Supplementary Fig. 4b).

Notably, both the permutation and automatic response detection analyses examine the responses at the level of individual units. Thus they may underestimate the actual timing of the population-level MTL anticipatory activity, which might have started earlier but did not reach significance at the unit level. In order to quantify the timing of unit activity at the population level, we ran the same permutation procedure, this time with the FR change percentage of all rivalry/replay-active units (see Methods). We found that group activity diverged from baseline 1072 ms prior to the report of transition onset to the preferred image during rivalry, compared to 455 ms during replay (Fig. 4a) —a ~600 ms difference (for rivalry vs. replay activity in the different subregions in the MTL, see Supplementary Fig. 5a). Note that here we separately analyzed rivalry-active and replay-active units, even though many units were active in both rivalry and replay. Supplementary Figs. 6 and 7 present the same analysis for the intersection and union of these two populations, respectively, with very similar results.

To further validate these results, we considered and rejected four possible confounds that could have biased the analysis. First, since we timed the activity by comparing the instantaneous FR against the baseline FR in each trial, the difference between rivalry and replay could have stemmed from a difference in baseline FR rather than actual timing difference. To rule out this possibility, we compared the baseline FR in rivalry and replay and found no difference between the conditions, both when inspecting only rivalry/replay-active units ($t$(39) = 0.11, $p$ = 0.91, two-tailed paired $t$-test), or all non-rivalrous condition responsive units ($t$(104) = 0.39, $p$ = 0.7, two-tailed paired $t$-test).

A second possible concern is that the earlier rivalry response reflects the higher statistical power in rivalry that results from the larger number of trials in this condition (2–5 rivalry vs. 1–2 replay blocks). To control for this concern, we repeated the analysis while equating the number of trials in the two conditions. A similar pattern of results was found (Supplementary Fig. 8).

A third concern is that the earlier activity relative to transition onset (end of dominance period) simply reflects the difference in dominance durations between rivalry and replay (see behavioral results above). This is less likely since the anticipatory activity was longer (i.e. started earlier) during rivalry, where dominance durations were actually shorter. Yet to make sure there is no relation between firing onset time and dominance duration, we computed the correlation between the trial-by-trial perceptual transition activity onset time (as determined by the trial-by-trial response detection analysis) for each unit, and the preceding dominance period duration. No significant correlation was found

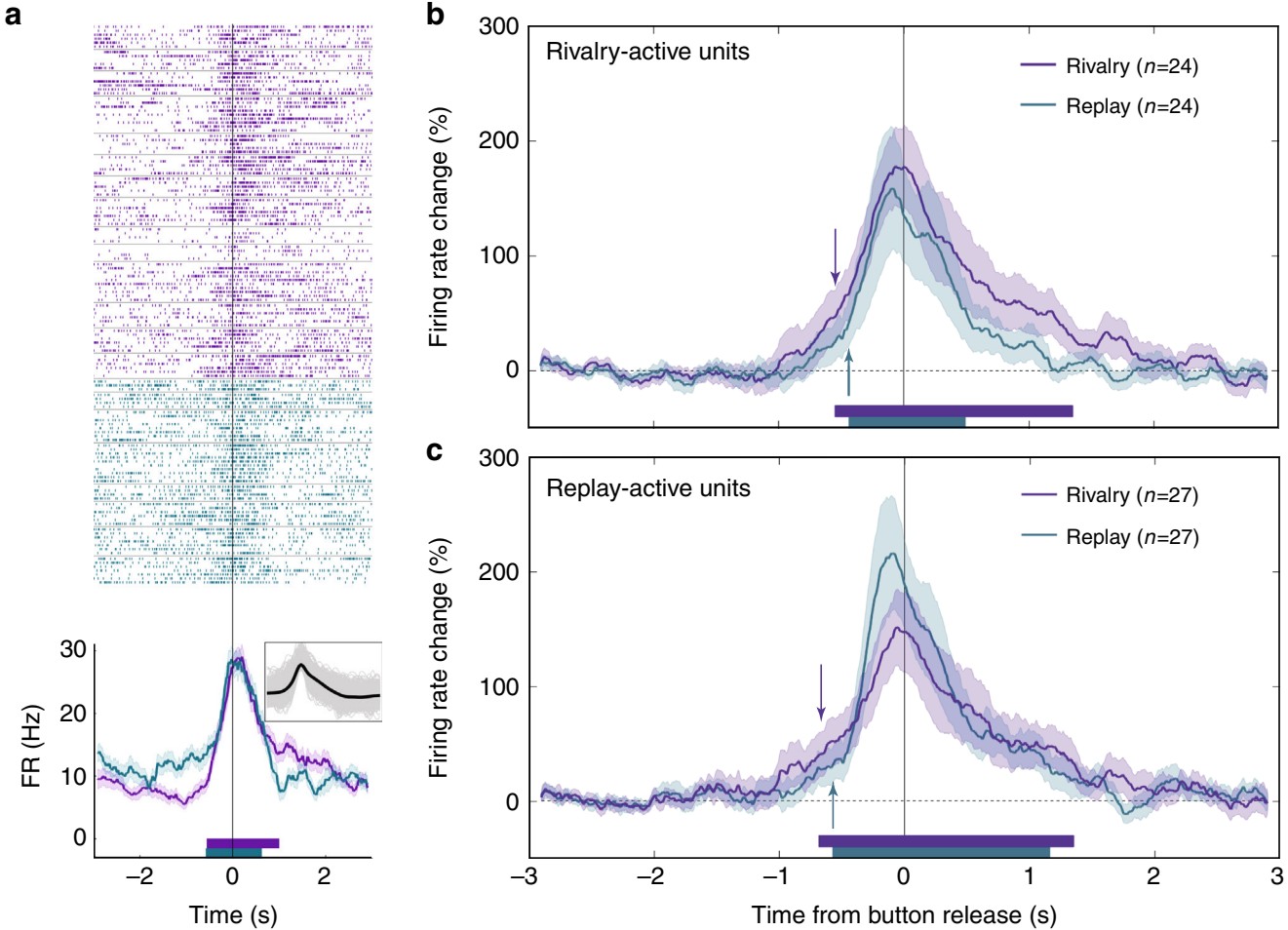

**Fig. 6** SMA activity during rivalry and replay. Convention as in Fig. 5. **a** Firing of an SMA multi-unit around the report of a perceptual transition onset during rivalry (purple) and replay (cyan). Note that this anticipatory activity was nonselective and accompanied the transition to any image. Average FR percent change across all rivalry-active units (**b**) and replay-active units (**c**). Note that here activity onset during rivalry and replay is highly similar

(all corrected $p$ values > 0.4; Simes correction for multiple comparisons[39]).

Finally, to test the robustness of these results and validate that they do not depend on a few outliers, we conducted a bootstrapping analysis, in which we repeated the analysis 10,000 times while sampling units randomly with replacement. The mean bootstrapped activity onset value was −1004 ± 160 ms and −545 ± 191 ms for rivalry and replay respectively (Supplementary Fig. 9). Note that the actual replay activity onset (−455 ms) was later than 99.9998% of the bootstrapped rivalry onsets, while the actual rivalry activity onset (−1072 ms) was earlier than 99.9826% of the bootstrapped replay onsets.

**Medial frontal activity precedes MTL activity.** We recorded from a total of 134 (54 single) anterior cingulate cortex (ACC) units in three patients and 50 (29 single) pre-supplementary motor area (preSMA) units in two patients (see Supplementary Table 1). These neurons did not show selective responses during the non-rivalrous condition. Inspired by fMRI reports of frontal involvement in perceptual transitions[15,16,21,23,24,40–42] (but see ref.[43]) we looked for units that changed their FR compared to baseline around the perceptual transitions, irrespective of image identity. Sixty such units (33%) were found in ACC/preSMA. Figure 5a depicts an exemplary ACC unit. To assess the activity onset difference between rivalry and replay at the population level, the same group-level permutation analysis used in MTL was

conducted. Here, the anticipatory activity of rivalry-active ACC/preSMA units ($n = 39$; defined as units that were active during rivalry at the unit level; 19 of them decreased their FR below baseline) started 1885 ms prior to the rivalry transition report, but only 131 ms prior to the replay transition report (Fig. 5b; for individual patient data see Supplementary Fig. 10a). In comparison, MTL group activity started only 1072 ms prior to rivalry transition report. This regional difference holds also for the subset of patients with responsive units in both regions (Supplementary Fig. 11), suggesting that this difference does not stem from differential response times between patients.

The anticipatory activity of replay-active ACC/pre-SMA units ($n = 26$; defined as units that were active during replay at the unit level) started 224 ms prior to the replay transition report, and 324 ms after the rivalry transition report (Fig. 5c; for individual patient data see Supplementary Fig. 10b).

We also recorded from 39 (9 single) supplementary motor area (SMA) units in one patient. Eighty-five percent of them were rivalry/replay-active (33 units; for an example see Fig. 6a; seven of them decreased their FR below baseline). Interestingly, population-level activity had similar timing in rivalry and replay: 547 ms and 436 ms prior to the report of perceptual transition during rivalry and replay, respectively, for rivalry-active units ($n = 24$; Fig. 6b), and 683 ms and 571 ms for replay-active units ($n = 27$; Fig. 6c). As these units are all from the same patient we cannot assess the generality of these findings.

## Discussion

This study reports human single unit correlates of internally driven changes in the content of visual awareness in the medial temporal and frontal lobes. Strikingly, we found early activity, which preceded the report of internally generated perceptual transitions by as much as 2 s in the ACC/preSMA and 1 s in the MTL. Activity in these regions started at least half a second later when the perceptual transitions were externally (retinal) driven during replay. These findings extend those from previous studies in several ways: first, they target neurons in higher level areas in the visual hierarchy compared with previous intracranial BR studies in monkeys that did not surpass the inferior temporal cortex[44]. Similarly, as opposed to previous monkey single unit studies of subjective perception that targeted lateral frontal areas[45–48], this study focused on neurons in medial frontal areas (specifically the preSMA and ACC). Second, the high spatio-temporal resolution of single unit recordings allowed us to extend previous fMRI BR studies[15,21,49,50] in closely tracking the time-courses of MTL vs. frontal activity during rivalry and replay. These comparisons showed that ACC/preSMA activity precedes MTL activity.

Furthermore, the results suggest that the temporal dynamics of rivalry is different from that of replay: while replay activity was strong and more centered around the report of perceptual transition onset, rivalry activity was prolonged and started earlier relative to the report. Most importantly, the use of BR allowed us to focus on internally driven changes in perception, in contrast to previous human intracranial studies that examined externally driven changes[1–3] (although see preliminary results from BR[51]). Thus, we were able to examine internal mechanisms that are involved in conscious perception, and—as opposed to the above studies which could only track the relations between MTL firing and stimulus onset—to determine the temporal relations between neuronal responses and subjective conscious perception, as reflected by patients' reports. Indeed, we found activity that substantially precedes the report of perceptual transitions.

It is important to note that here we focused on the report of the beginning of the transition period as opposed to most previous studies of BR that focused on the beginning of the dominance period (e.g. refs. [10,11,49]). In our opinion, the former is more meaningful because it marks the emergence of the new image into consciousness, while the latter marks the end of the transition period, during which the new image was already partially perceived. Note that for the patients, determining the exact point in which an image became dominant (e.g., differentiating between a state in which the image is 90% dominant to 100% dominant), was more difficult than determining when a change—irrespective of its magnitude—first occurred. For completeness, Supplementary Figs. 12 and 13 present the responses aligned to the report of the beginning of the dominance period for MTL and frontal units, respectively (Supplementary Fig. 5b presents this data for each MTL subregion separately). As expected, when MTL activations are locked to dominance onset, they begin earlier during both the rivalry and replay conditions, and the difference in activity onset between the conditions is even larger. In the ACC/preSMA, fewer units were active around the dominance event than around the transition event (21 vs. 39 units, respectively) and the responses were weaker, attesting to the relevance of this frontal activity to the transition itself (or the mechanisms which trigger it); however the difference between the conditions is still well preserved.

These early activations, however, could potentially be attributed to long post-perceptual processes, which take place after the transition has occurred, yet prior to report, especially in a clinical environment. This interpretation does not seem likely given how early these responses were, especially in the ACC/preSMA, and

given that replay responses started much later. Yet one could still claim that post-perceptual processes are different in the two conditions, and might be shorter in replay. This would suggest that the perceptual transition actually occurred earlier during rivalry than during replay, indexed by firing onset, but it took longer to report it during rivalry. Admittedly, even though we used a matched-duration replay, it was still not perceptually identical to rivalry experience, which is distinctive and often fragmented and wave-like[52] and accordingly difficult to simulate. While an experienced observer, if asked to, could probably differentiate between the conditions, patients had no prior experience with rivalry, hence unlikely to spot the difference, and were not informed about the two distinct presentation methods. Indeed, the behavioral time courses of rivalry and replay are quite close (see an example in Fig. 1d, and the relatively similar gamma distributions in Supplementary Fig. 1). In addition, if the rivalry-replay activity onset differences found in the MTL and ACC/preSMA only stem from differential post-perceptual processes, they should affect all neuronal responses. But several MTL units responded at the same time in the two conditions, or even earlier in replay (see color traces in Supplementary Fig. 6b,c). Similarly, in one patient from which we recorded from SMA, activation profiles were almost identical for rivalry and replay. Thus, we do not think that a difference in post-perceptual processes can explain the results.

The early ACC/preSMA activity—preceding even MTL neurons—is especially surprising given several recent studies that questioned frontal involvement in BR, in particular, and in conscious experience, in general[29]. These studies showed that when the duration of perceptual events in rivalry and replay is matched[20], or when subjects are not required to report[18,19,53], the difference in frontal activity between rivalry and replay is substantially reduced or even eliminated. The matched duration criticism does not apply here, since we used gradual replay transitions, matched in duration to the rivalry behavioral time course. While we cannot rule out that ACC/preSMA activity relates to report, this does not seem likely given that the response starts 2 s before the report, and precedes MTL activity. Notably, if ACC/preSMA activity relates only to the report, we would expect it to follow MTL activity, held to correlate with perception[1,2].

This suggests that ACC/preSMA might have a role in settling the ongoing conflict between the two rivaling images throughout rivalry, in line with ACC and preSMA suggested role in conflict monitoring[54–57] and in executive functions[58]. Notably, this should not be taken as evidence that these areas are necessarily the earliest involved in the perceptual switch; it could be, for instance, that the switch is triggered by a subcortical stochastic oscillator[59,60] (e.g. locus coeruleus[61] which is the source of nor-adrenaline in the forebrain[62] and is involved in attention, or raphe nuclei, responsible for the release of serotonin[63], whose levels have been shown to affect rivalry alternations[64]) that projects to these frontal structures. While medial frontal activation has been reported in a few fMRI studies of bistable perception[15,20,21,23,24], most of these studies found activations in lateral prefrontal regions. The region that is most consistently implicated in imaging studies (also after controlling for the matched duration[16] and no-report criticism[19,65,66]) is the inferior frontal cortex (IFC). Intriguingly, both the preSMA and the ACC are anatomically connected to the IFC[67,68] and activity in these regions has been shown to actually precede that of IFC in bistable and conflict monitoring situations[23,69]. Notably, dissociation between single-unit and imaging findings has been reported in the context of BR[70], and is not uncommon in general[71,72], specifically in the prefrontal cortex[30]. For example, the frontal eye field was implicated in bistable perception in single-unit studies[45–47], but

not in the above-mentioned imaging studies. Thus, the under-representation of preSMA and ACC in imaging studies of BR does not necessarily contradict our findings.

What conclusions can be drawn from these results, then, regarding the mechanisms of BR, and—more generally—about the processes that lead to the emergence of a new percept? Most computational models of BR emphasize competitive inhibitory interactions at multiple neural sites, as well as feedback connections[73] and attentional modulations, mediated by signals from late to early visual areas and/or signals from frontoparietal areas[74]. Our results are in general agreement with such models, with the MTL either sending feedback signals to bias the competition that takes place in lower level areas or being the locus of competition itself, especially for high-level rivaling stimuli as used here. In this regard, it is interesting to note that MTL activity actually displayed competitional dynamics, being more gradual and prolonged in rivalry as compared to replay. One might have speculated that the decreasing FR responses in MTL neurons represent inhibition or some form of adaptation, yet this seems less likely because these responses appear also in the non-rivalrous condition, where there is no competition. Rather, the neurons with negative responses are part of an MTL network in which some neurons respond by increasing their FR while other respond by decreasing it. A typical activity in any part of the network might reinstate the activity in other parts of the network and lead to the emergence of the represented concept. The earlier frontal activity—whether positive or negative—can also serve as attentional signal that biases the competition in MTL or lower-level regions. However as our study was limited to specific areas dictated by clinical considerations, and does not include low-level visual areas nor most of frontoparietal cortex, we cannot provide a full description of the underlying mechanisms.

Taken together, our findings suggest that internal changes in the content of perception may be influenced by early activity in a cortical network that includes the ACC and preSMA followed by MTL activity that presumably leads to the perceptual change. The abstract, conceptual representations of these MTL units, previously shown to be reinstated prior to the emergence of a memory[14], may be likewise involved in the internal emergence of a percept, as our findings suggest. On a more theoretical level, these processes may be quite similar: during free recall, the pattern of MTL activity that accompanied the actual experience is internally regenerated, leading to the perceptual state of re-experience, which is the recall event. Similarly, during BR the pattern of MTL activity might lead to the internal generation of a new perceptual state.

## Methods

**Patients and recordings**. The data were collected in 20 recording sessions in 9 patients (mean age 39, range 19–50, 5 females) with pharmacologically intractable epilepsy. Extensive noninvasive monitoring did not yield concordant data corresponding to a single resectable epileptogenic focus. Therefore, they were implanted with chronic depth electrodes for 7–10 days to determine the seizure focus for possible surgical resection[31]. Electrode locations were based exclusively on clinical criteria and were verified by postimplant computer tomography (CT) coregistered to preimplant magnetic resonance imaging (MRI). Each electrode consisted of a flexible polyurethane probe containing nine 40-µm platinum–iridium microwires protruding ~4 mm into the tissue beyond the tip of the probe. Eight microwires were active recording channels and referenced to the ninth, lower impedance, microwire. The differential signal from the microwires was amplified by using a 128-channel Blackrock™ system, filtered between 0.3 and 7500 Hz and sampled at 30 kHz. Here we report data from sites in the hippocampus, amygdala, entorhinal cortex, parahippocampal gyrus, anterior cingulate cortex, presupplementary motor area and supplementary motor area. 33 ± 19 units per session were recorded in these areas. All sessions were conducted at the patients' quiet bedside. All studies conformed to the guidelines of the Medical Institutional Review Boards at University of California at Los Angeles and Tel-Aviv Sourasky Medical Center, and all patients provided written consent forms.

**Unit identification and spike sorting**. Spike detection and sorting was applied to the continuous recordings by using the well-established "wave-clus" software[75]. Briefly, extracellular microwire recordings were high-pass filtered above 300 Hz, a threshold was set at 5 SD above the median noise level and detected events were sorted using superparamagnetic clustering. After sorting, the clusters were classified into noise, single-unit or multi-unit based on: (i) the spike shape and its variance; (ii) the ratio between the spike peak value and the noise level; (iii) the interspike interval (ISI) distribution of each cluster; and (iv) the presence of a refractory period for the single units, i.e., less than 1% spikes within 3 ms ISI[32]. Forty-one percent of the units were classified as single units (see Supplementary Table S1). On average 1.4 ± 0.6 units were identified per wire. We computed several spike sorting quality measures for all identified units (Supplementary Fig. 2): (a) percentage of ISIs below 3 ms was 0.84% ± 1.83% (0.35% ± 0.42% for units classified as single units); (b) the ratio between the peak-to-peak amplitude of the mean waveform of each cluster and the SD of the residuals was 7.9 ± 3.3 (SNR[76]; 10.1 ± 3.7 for single units); (c) the median L-ratio[77], which is the amount of contamination of a given cluster based on the Mahalanobis distance of spikes not included in the cluster from the center of the cluster, divided by the total number of spikes in the cluster, was 0.03 (SD = 0.69).

**Stimuli and procedure**. Visual stimuli were generated by MATLAB with the Psychophysics Toolbox extension[78], running on a 15-inch Apple MacBook Pro laptop. Stimuli were presented on the laptop screen at 60 Hz and screen resolution of 1440 × 900. Responses were collected using a Logitech F310 gamepad.

**Selectivity screening session**. In a first recording session, usually done early in the morning, a large number of images (107 ± 25) of famous people, landmarks, animals, objects and family members were presented to the patient. This set was composed based on patient's preferences. 300 × 300 pixels images (5° visual angle) were presented for 1 s followed by a blank screen of 0.5–1 s, and repeated 6–8 times in a pseudorandom order, while patients were engaged in simple discrimination tasks (e.g. person/other, building/other, man/woman etc.). Images that elicited the strongest responses in the screening session (using the same procedure as in ref. [38]) were selected for use in the BR session that took place a few hours later. Data from the selectivity screening session is not presented here. Importantly, the BR session (see below) included a non-rivalrous condition in which images were presented normally to both eyes. Units were selected for rivalry/replay analysis based on the results of this condition and not based on the screening session results.

**Binocular presentation**. During the BR session, the visual stimulation to the left and right eyes was independently controlled using one of the following two methods[79]:

(1) Red-blue goggles (n = 6 patients): The two visual streams were presented in either red or blue at the center of the screen. Each lens passes only one of the streams, so that the two streams fall on corresponding retinal locations of the two eyes.

(2) Mirror stereoscope (n = 3 patients: patients 2–4 in Supplementary Table 1; one of the five sessions of patient 4 was with red-blue goggles): patients viewed the screen via an adjustable mirror stereoscope (SA200LT, Stereo Aids, Australia www.stereoaids.com.au): Left and right eyes' visual streams were presented at different horizontal locations of the screen and were projected on corresponding locations of the retinae of the two eyes using the stereoscope.

Identical vergence cues (black and white dashed frames around the images, a 440 × 440 pixels cross (7° visual angle) behind the frames, and a 20 × 20 pixels fixation cross (0.3°) at the center of the images) were presented to both eyes 1 s before the beginning and throughout each of the conditions of the BR session, to ensure binocular fusion.

Note that the mirror stereoscope, used in the first three patients, was cumbersome to use in the clinical setting. Therefore we switched to the red-blue goggles that were more convenient and familiar to patients. Notably, the separation between the two eyes might be incomplete with the red-blue goggles, yet that actually makes it more difficult to find changes in firing locked to the perceptual transitions.

**Binocular rivalry session**. Based on time restrictions with the patient, a variable number of image pairs (M = 2.70 ± 0.73) were used in each session. Images that elicited the strongest responses in the screening session were usually paired with images that did not elicit a response in the same units. Before recording, patients were carefully instructed with the details of the task and were trained with a demo of rivalry, where transitions physically occurred on the screen, so that the experimenter could confirm that they execute the task well. Then patients wore red-blue goggles (or viewed the screen via a stereoscope, see "Binocular presentation" section above) and completed one or more repetitions of the following three conditions for each pair of images:

Non-rivalrous condition: A slide informing the patient on the assignment of one button (either left or right arrows of the gamepad) to each of the two images was presented. Patients were asked to press the assigned button for each image appearing on screen. Each of the two images (5° visual angle) was presented binocularly to both eyes (hence no binocular conflict; Fig. 1a) 8-10 times,

for 1 s or longer, until a correct response was made. Order of presentation was pseudo-random, and one-second interleaving blanks were used between images. To ensure correct button assignment, the non-rivalrous condition was stopped only after 8–10 successive correct responses to each image in the non-rivalrous condition that preceded the first rivalry block, and four correct responses in subsequent blocks.

Rivalry condition: The two images (5° visual angle) were presented simultaneously one to each eye (see "Binocular presentation" section above) for either 90 or 120 s. This type of presentation creates BR, in which each image dominates conscious perception for a certain period while the other image is perceptually suppressed. These dominance and suppression periods reverse irregularly, interleaved by periods of mixed percept (transition/piecemeal periods; Fig. 1b). Note that the physical stimulus is constant hence the perceptual transitions are internally driven. Patients were asked to report perceptual dominance onset of each image by pressing and holding the assigned button for that image, and to immediately release that button as soon as something in the image starts to change (dominance offset/ transition onset[11]). This scheme of reports provided four behavioral events: image1 dominance onset (button A press); image2 dominance onset (button B press); emergence of image1 = image2 dominance offset = image2→image1 transition onset (button B release); and emergence of image2 = image 1 dominance offset = image1→image2 transition onset (button A release). To avoid unbalanced duration due to ocular dominance the two images were switched between the eyes in the middle of the rivalry condition, by linearly increasing the transparency of the current image in each eye from zero to 100% while linearly decreasing the transparency of the other image from 100% to zero over the course of 1 s. Additionally, two catch trials, in which the same image was presented to both eyes for 1 s, were included. The transparency of one image was linearly ramped up to 100% while the transparency of the other image was ramped down to zero over the course of 1 s before and after the catch trial. Eye-switching and catch trials periods and the following 1 s after these periods were not included in the rivalry analysis.

Replay condition: The four types of reports from the rivalry condition were used to create a matched-duration replay condition that immediately followed the rivalry condition (Fig. 1c). In the replay condition, the same stimulus was always presented to both eyes. During dominance periods of each image (defined as the time between press and release of the corresponding button), that image was presented to both eyes. During transition/piecemeal periods (defined as the time between the release of one button to the press of the other button), the transparency of the previous image was linearly ramped up to 100% while the transparency of the next image was linearly ramped down to zero in both eyes. During incomplete transition periods, in which a button release was followed by the same button press, the transparency of the dominant image was linearly ramped up to 50% while the transparency of the other image was linearly ramped down to 50% over the first half of that piecemeal period, and then the transparency of the dominant image was ramped down back to zero while the transparency of the other image was ramped up back to 100% over the second half of that period. The replay stimulation was designed to externally generate a perception that would closely mimic rivalry, thus resulting in a matched sequence of motor responses in the two conditions (see for example, Fig. 1d). Patients' task was identical and patients were not informed that this condition is different from the rivalry condition. The replay condition was included for only one (usually the first) or two repetitions of the rivalry condition.

**Selectivity analysis**. The data from the non-rivalrous condition was used to define selective responses in MTL units. For each image, FR was calculated in three 250-ms bins starting 150 ms after image onset and ending 100 ms before image offset. FR outliers (>2 standard deviations above/below the mean) were discarded. For each bin, FR in all trials were compared to the FRs in a 250-ms window before all non-rivalrous image presentations (baseline FR) by means of a Mann−Whitney $U$ test, using the Simes correction for multiple comparisons[39] and applying a conservative significance threshold of $p = 0.005$[38]. This procedure identified both positive (i.e. increases above baseline FR; 72% of responses) and negative (i.e. decreases below baseline FR; 28% of responses) responses. Only MTL units that responded to one or more of the images in the non-rivalrous condition were included in the subsequent activity onset analysis. For each image pair, if a unit responded to only one of the images, this image was considered the "preferred image" for this unit, while the second image was the "non-preferred image". If a unit responded to both images, the image that elicited the response with the lower $p$value, or with the same $p$value but in more bins, was considered the "preferred image", and the other one was considered "non-preferred".

**Critical events for MTL and frontal units**. Two critical events were analyzed for each unit: perceptual transition onset (button release) during rivalry and replay. For MTL units, this pertained to the preferred image only, and analyzed separately for each image pair, while for frontal units, where there was no selectivity to specific images—all transition events were analyzed together, across all images of all pairs. The "Individual unit activity onset analysis" (see below) was focused on these events. Note that all transition onsets to the unit's preferred image, whether complete or incomplete (i.e. that do not lead to full dominance; 24%), were taken into account. For MTL units, only responsive units from the non-rivalrous

condition were included. For frontal units, all recorded units were subject to this analysis. Supplementary Figs. 12 and 13 show the same analysis for perceptual dominance onset events (button press).

**Individual unit activity onset analysis**. To identify activity that significantly differ from baseline and determine its onset, a permutation test was conducted with cluster-based multiple comparison correction across time points[80]: at each time point in the [−1500,1000 ms] window around the critical event, the instantaneous FR (iFR; calculated with a sliding square window of 200 ms) was compared to the baseline FR (mean iFR during [−3000,−2000ms] window; first 100 ms were discarded due to edge effect of the square window smoothing) over trials, using a paired two-tailed $t$-test. Temporal clusters of significant activity were defined as consecutive significant timepoints ($p < 0.05$) with a maximal gap of 100 ms, and were assigned a cluster-level statistic corresponding to the sum of the $t$values of the time points belonging to that cluster ($t$-total). The distribution of maximal absolute cluster-level statistics obtained by chance was estimated by repeating the analysis 1000 times with randomly rearranging the spikes in a [−3000,3000 ms] window of each trial while preserving the ISI distribution of that trial. Only clusters with absolute $t$-total in the top 5% of this distribution ($p < 0.05$) were considered significant. This method allowed us to both detect positive or negative activity (i.e. significant increase or decrease relative to baseline FR, respectively) and measure its onset. Out of these significant clusters, only the one that was closest to the critical event ($t = 0$) was included in subsequent analyses. Clusters that started more than 500 ms after the critical event were discarded. Note that the baseline time window was not completely clean—for 15% of the perceptual transitions trials, the [−3000, −2000 ms] time window included the previous dominance onset event, and thus might have also included significance changes in FR. This results from the continuous nature of the paradigm and critically only makes it more difficult to detect an effect.

**Population-level activity onset analysis**. The above units were further analyzed at the population level. Here instead of conducting a $t$-test over trials at each timepoint, we averaged the iFR across trials for each trace (a unit response to a certain image-pair) and conducted a $t$-test over traces at each timepoint. As different traces had different baseline FR and either positive or negative activations (i.e. increased/decreased FR relative to baseline FR), we looked at absolute iFR percent change relative to baseline. To do so we normalized the iFR time-course to 0–1 range, and inverted the timecourse (1-timecourse) of traces that had a negative $t$-total in the individual unit activity onset analysis. The iFR percent change was calculated as the ratio of the difference (iFR − baseline FR), and baseline FR. The permuted data ($n = 1000$) was generated by separately shuffling the spikes of each unit and each trial while preserving the ISI distribution. Traces for which the iFR timecourse was inverted in the real data were inverted also in the permuted data. Three units with FR lower than 0.5 Hz were discarded from the analysis.

**Bayes factor analysis**. We calculated the Bayes factor (BF), defined as the ratio of the probability of observing the data given H0 and the probability of observing the data given H1, using JASP (Version 0.8.5; JASP team, 2017). We adopted the convention that a BF less than 0.1 implies strong evidence for lack of an effect (that is, the data are ten times more likely to be observed given H0 than given H1), a BF between 0.1 and 0.33 provides moderate evidence for lack of an effect, BF between 0.33 and 3 suggests insensitivity of the data, BF between 3 and 10 denotes moderate evidence for the presence of an effect (i.e., H1), BF greater than 10 implies strong evidence and BF greater than 100 suggests extreme evidence for the presence of an effect[37].

**Automatic trial-by-trial response detection analysis**. Trial-by-trial activity onset times relative to the perceptual transition reports were determined by Poisson spike train analysis (Hanes et al., 1995)[81]. For this procedure, the ISIs of a given unit are processed continuously over the [−1500, +1500 ms] window around the report, and the onset of a spike train is detected based on its deviation from a baseline exponential distribution of ISIs (i.e. an exponential distribution with $\lambda = 1/$mean FR in [−3000, +3000 ms] window across all trials). Only spike train onsets <0 (i.e. earlier than the motor response) were considered. Activity onset time in rivalry and replay was determined for each trace as the median onset time for this condition, with the constraint that significant activity onset was recognized in at least 40% of the condition trials for this trace. An example of this analysis appears in Supplementary Fig. 4a. Note that this analysis can only recognize increases in FR relative to baseline, and therefore was not used on traces with decreased FR response.

**Bootstrapping analysis**. The bootstrapping analysis was aimed at assessing the robustness of the effects found at the population-level activity onset analysis. This analysis was repeated 10,000 times, so that in each iteration a random sample of 36/26 traces (corresponding to the number of MTL active traces in rivalry/replay) was selected with replacement (allowing repetition). The population permutation test described above was conducted for each sample, and the onset was determined both for replay and for rivalry. The bootstrapped onset distributions were plotted (Supplementary Fig. 9), and their means were calculated. Finally, we calculated the

chances of obtaining the actual rivalry onset under the bootstrapped replay distribution, and the chances of obtaining the actual replay onset under the bootstrapped rivalry distribution.

**Data availability**. The data that support the findings of this study are available from the corresponding author upon reasonable request.

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

## Acknowledgements

We thank the patients for their participation; Rafi Malach, Yuval Nir, Tal Golan, Tomer Livne, Dan Biderman, and Natalie Biderman for discussions; Amit Marmelshtein for code; Michelle Tran, Deena Pourshaban, and Mariana Holliday for assistance with data acquisition; Eric Behnke and Tony Fields for technical assistance; Brooke Salaz for administrative help. C.K. thanks the Allen Institute founder, Paul G. Allen, for his vision, encouragement, and support. This work was supported by grants from the National Institute of Neurological Disorders and Stroke (NINDS no. R01NS033221 and R01NS084017) and Mathers foundation, fellowships from the Human Frontiers Research Program (HFSP), European Union's Horizon 2020 Marie Skłodowska-Curie Actions (MSCA; under grant agreement no 659759 and 659765), the European Molecular Biology Organization (EMBO), the Israel National Postdoctoral Award Program for Advancing Women in Science, and the L'Oréal-UNESCO For Women in Science program.

## Author contributions

H.G.-S., C.K., and I.F. designed the experiment. I.F. performed the surgeries. M.R.H. contributed the pre-screening presentation and analysis code. H.G.-S. implemented the paradigm and collected the data. H.G.-S. and L.M. analyzed the data. H.G.-S., L.M., and I. F. wrote the manuscript. H.G.-S., L.M., C.K. and I.F. discussed the results and commented on the manuscript. All authors read and approved the final manuscript.

## Additional information

**Competing interests:** The authors declare no competing interests.

