## [Peer Review File(PDF 6583 kb) · Nature Communications]

Reviewers' comments:

Reviewer #1 (Remarks to the Author):

The manuscript reports the activity of single neurons recorded in the medial temporal and medial frontal cortex of patients undergoing epilepsy surgery. The participants were asked to report binocular rivalry alternations while neural activity was recorded. The authors report that the beginning of a dominance phase was preceded by modulation of activity in medial frontal and medial temporal cortex. The activity in medial frontal preceded the dominance transition by as much as 2000 ms, and that in medial temporal, by as much as 1000 ms. Modulation was contrasted with a control condition of experimenter imposed image transitions.

The results are hard-won and unique. Major observations seem reliable. The findings should be of interest to diverse investigators and scholars. However, enthusiasm for the manuscript is diminished by the following issues:

(A) A proliferation of undefined and un-measured (and un-measurable) mental terms creates more confusion than clarity. Examples include Line 22-23, "conscious perception was manipulated using two paradigms..." – What would unconscious perception be? Line 299-300, "These early activations ... could potentially be attributed to long decision processes..." – What is a decision process? How would one know whether or not it is happening?

(B) Readily acknowledging the challenges involved in obtaining such data, it must be wondered whether the sample size is sufficient to support the conclusions. From MTL the dataset consists of just 31 units were active during rivalry or replay, but only 19 were active during both conditions. Meanwhile, 10 were active exclusively in rivalry and only 2 exclusively in replay. These are small numbers, but it is also curious that 12 of the neurons distinguished the stimulus presentation condition. This indicates that something more than "rivalry" and "replay" is being noticed. The medial frontal sample consists of 39 units. Other peculiarities can be noticed. For example, in the bottom raster of Fig 2c during rivalry, a clear change of discharge rate precedes the synchronization time by ~800 ms. Such alignment can also be observed in the top raster. How common was this? More generally, methods have been developed to quantify modulation in single spike trains. The results would be enhanced by application of such methods.

(C) The mechanisms mediated by the modulations are entirely unclear. The authors emphasize one preferred interpretation, but viable alternatives are not considered or not ruled out conclusively. The authors state (line 343ff) that "internal changes in the content of perception may be heralded by early activity in a cortical network that includes the ACC and preSMA". How does "heralding" work? What is the computational theory of "heralding"? It is a poetic and evocative word, but it has no scientific content.

(D) The novelty of the findings and their relation to previous observations is less clear than

it could be. The authors claim their results can resolve “conflicting findings” in a “debate ... focused on the role of frontal areas in internally driven perceptual alternations”. This claim is challenged by the fact that the authors’ data is from a part of frontal lobe different from those described in previous studies. Also, the authors seem to overlook monkey single unit recordings addressing neural correlates of subjective awareness in the frontal eye field (Thompson & Schall 1999, 2000; Libedinsky & Livingstone 2011).

(E) Ultimately, the theoretical insights offered by the findings are minimal. The Discussion is a meandering summary of the results and the authors’ preferred interpretation. No alternative hypotheses are seriously excluded. The take-away stated by the authors in the Abstract is simply a restatement of the observation.

OTHER COMMENTS

(1) Can the participants distinguish actual rivalry from replay alternations? Do they know which they are experiencing? In 2AFC condition, could they (or anyone else) distinguish the two stimulus conditions? If so, the reliability of the control condition can be questioned.

(2) The logic of measuring the beginning of a transition is clear, but the practice can be problematic. In measuring the beginning of the transition period, how did the authors instruct or train the participants to judge this reliably? This is particularly problematic because the displays subtended 5 degree of visual angle, which supports piecemeal rivalry, especially with complex object and faces as used.

(3) On lines 338ff the authors argue that medial frontal activation was not observed in fMRI studies of bistable perception because they found both decreases and increases of discharge rate, which would average out in an fMRI voxel. This argument seemed flawed. Both increases and decreases of discharge rate entail metabolic processes utilizing oxygen, so why wouldn’t BOLD increase?

(4) The quality of unit isolation was not documented.

(5) Typographical errors are sprinkled through the manuscript.

Reviewer #2 (Remarks to the Author):

This is an interesting manuscript that reports a technically complex experiment with a simple and potentially important finding. The experiment involved intracranial recording of single neurons in the human medial temporal lobe, anterior cingulate or pre-SMA while participants reported binocular rivalry or a matched replay condition. Participants’ reporting included explicit tracking of both the start and finish of the transition period between each monocular image (or binocular replay transition). This allowed the authors to measure and analyse single unit firing time-locked to the start of the rivalry (or replay) transitions.

Major comments

1. Line 137 (and elsewhere) the use of 'emergence' is confusing, as if I have understood correctly this is the time point at which the participant judges that the dominant image is starting to transition. A second button is then subsequently pressed after this button release when the other monocular image is fully visible. 'Emergence' might properly be used to refer to this second timepoint at the end of the transition period. It would be helpful to have figure 1 annotated to clearly describe which time point was used for averaging of responses, and perhaps drop the use of 'emergence' which is ambiguous concerning whether it is the start or end of emergence.

2. One of the challenges of interpreting anticipatory activity is that the process of rivalry is continuous. For the ACC/pre-SMA activations in particular, is it possible that the premonitory activity for a button release is in fact related to the previous button press timing?

3. It is notable that the study found differences in the timing of anticipatory activity between rivalry and replay, but also found differences in how transition periods were reported in rivalry and replay and differences in the dominance durations between rivalry and replay (despite the efforts to match these conditions). Could the differences in anticipatory activity be related to these timing differences? One way to examine this – if possible – might be to examine correlations on a trial-by-trial basis between the timing of the anticipatory activity and the (trial-by-trial) measurement of the transition period (or immediately preceding dominance period) duration.

4. Line 337-338. Scholarship concerning the human BR studies needs improving. I had a look at reference 13 (Lumer et al) and could not clearly identify ACC activity in Figure 2 and this is reported in Table 2. This is the opposite of how this reference is cited. Similarly, reference 14 Figure 3 describes activity in left SMA which is quite close to pre-SMA. I suggest this paragraph is rewritten to accurately report the references.

5. Line 379. It would be helpful to have a flow diagram to understand how selective the ultimate findings were. How many units were recorded, then how many were classified as single units, then how many were selective and proceeded to the BR sessions, then how many were actually entered into the analysis. Such a diagram/table would help the reader understand how representative the neurons that are reported might be with respect to the overall populations recorded

6. Lines 398-410. Two different methods of binocular rivalry presentation were employed. As the separation between the two visual streams is likely to be complete with the mirror stereoscope but incomplete with the red-blue goggles (which will also attenuate the images somewhat) it would be helpful to confirm whether there is any difference in the key findings when the data are split according to visual stimulation technique.

Reviewer #3 (Remarks to the Author):

This manuscript, 'Human single neuron activity precedes the emergence of conscious perception', describes the authors efforts to provide evidence of single neuron involvement in subjective perception during binocular rivalry. The recordings were obtained from microwires implanted in neurosurgical patients who were being monitored for epilepsy. The authors were specifically interested in demonstrating changes in spiking activity in the medial temporal lobe (MTL) and in frontal lobe structures (the ACC and SMA) prior to the moment when individuals consciously perceive a change in an image.

This builds upon previous work demonstrating changes in neural activity along the visual processing pathways during binocular rivalry. The authors extend this work in several important ways. First, they demonstrate that changes in MTL neuronal activity precedes perception. The MTL in many ways sits at the top of the visual hierarchy, and so this work suggests . Second, they provide a novel control condition that is matched for the duration of transitions, which they call replay, and show that the timing of neural responses are slower in this replay condition. This suggest that internally driven transitions involve earlier neuronal activity. And third, they provide novel evidence that neurons within the ACC also exhibit early responses to internally generated transitions. In this sense, then, this work collectively begins to address the question of how humans create a subjective conscious experience.

The manuscript is well written and clear, and I believe the results are compelling. The statistical methods used to demonstrate changes in firing rates from baseline are appropriate and well done. I believe that the central interpretation, that MTL neuronal activity precedes subjective transitions is well supported. I also believe that the same claim about ACC neurons is also supported. I have the following concerns that, if addressed, could improve the overall manuscript:

Major Concerns

Single unit recordings from the human brain are quite challenging, and one of the issues to consider is the fact that a given unit recorded on an electrode during one experimental session may not be present during a separate session recorded even just hours later. In their experimental setup, each test of BR is preceded by a separate experimental session to identify unit responses to different images. Isn't there a concern that the units identified during these initial localization sessions would be different than the ones used in the BR experiments? How do the authors account for this?

In identifying units that are responsive to individual images, and therefore to determine which image is the preferred image for each pair, it appears that the authors treat both increasing and decreasing responses in the same way. This is relevant when looking at both the individual and population histograms, since the authors then invert any negative changes in firing rates to determine population level responses (Fig 4). However, it is unclear how to interpret this and how often this occurs. I think that that one interpretation is that there are changes in neural activity that precede transitions regardless of direction,

but it would be important to clarify and demonstrate how many of these responses are actually decreases in spiking activity. Also, it would be helpful if the authors could comment on how such selective decreases in activity might play a role in such conscious perception.

There is the possibility that an image is dominant, and then loses dominance only to immediately regain it. I think most of the text is written to consider only the possibility that the dominant image switches. But what happens in these cases when dominance is lost, only to be regained by the same image immediately afterwards? This would be helpful to know, as the question would be whether the same neurons become active once again in this situation. For example, suppose a non-preferred image is dominant, and suppose that a button release is initiated, followed by the same button press later (indicating that the same non-preferred image is dominant again). During that transition, are there still changes in spiking activity related to the preferred image? Perhaps. However, in the converse situation, when the preferred image is dominant, then transitions, and then is dominant again, I might expect not to see changes in spiking activity associated with that non-preferred image during the transition.

The statistical testing comparing rivalry to replay activity at the level of the population responses is sound, and the bootstrapping analysis provides some validation to these results. However, it would be important to know whether this effect is driven by only a small subset of neurons within only one or two individuals, or whether it is more evenly distributed. Table 1 would suggest that most of the data are captured from only a small number of individuals. As such, I would suggest that the authors perform these statistical comparisons both at the level of individual units (they could use a permutation test comparing rivalry to replay conditions), and for units aggregated within individual subjects, and report how frequently this effect arises.

The results describing neuronal activity in the ACC are also compelling, and I think the interpretation that neuronal activity in this region precedes these transitions is also well supported. However, I do think that one of the central claims of the manuscript, that ACC activity precedes MTL activity is hard to justify, primarily because there is no direct comparison within an individual patient between ACC and MTL activity. As such, there is no direct statistical test between them. I think the authors can still support most of the claims in the manuscript, but I think they should consider tempering the specific claim that ACC activity precedes MTL activity.

Are there any differences in MTL neuronal activity between the different subregions?

The authors raise the issue of ocular dominance, and state that the subjects' percepts were equally distributed between the two images. However, while the centers of the gamma distributions used to illustrate the frequency of relative image dominance lie around 1, the distributions clearly have longer tails to the right. This is clear in the distribution of transitions shown in Fig 1c, where most of the percepts are for the image of the snakes. I don't think this affects the main conclusions of the paper, but it would be worth discussing the issue of ocular dominance in the context of these results.

Reviewers' comments (our responses in blue):

Reviewer #1 (Remarks to the Author):

The manuscript reports the activity of single neurons recorded in the medial temporal and medial frontal cortex of patients undergoing epilepsy surgery. The participants were asked to report binocular rivalry alternations while neural activity was recorded. The authors report that the beginning of a dominance phase was preceded by modulation of activity in medial frontal and medial temporal cortex. The activity in medial frontal preceded the dominance transition by as much as 2000 ms, and that in medial temporal, by as much as 1000 ms. Modulation was contrasted with a control condition of experimenter imposed image transitions.

The results are hard-won and unique. Major observations seem reliable. The findings should be of interest to diverse investigators and scholars. However, enthusiasm for the manuscript is diminished by the following issues:

>> We thank the reviewer for recognizing the uniqueness of our findings and their merits, as well as for the important comments and suggestions that we address below.

(A) A proliferation of undefined and un-measured (and un-measurable) mental terms creates more confusion than clarity. Examples include Line 22-23, “conscious perception was manipulated using two paradigms...” – What would unconscious perception be? Line 299-300, “These early activations ... could potentially be attributed to long decision processes...” – What

is a decision process? How would one know whether or not it is happening?

>> We agree that some of the terms in the manuscript are not easy to define. In many ways, this problem is inherent to the field of consciousness research, given the complexity of the studied phenomenon. Following the reviewer's comment we tried to minimize the use of ambiguous terms (e.g., the term "gives rise" was removed; the term "perceptual changes" was replaced by the more accurate term "perceptual alternations". Also, in most instances in the text we replaced the word "emergence" with "perceptual transition"). Specifically regarding the terms noted by the reviewer, we changed "decision process" to "post-perceptual processes" which is more accurate and reflect an ongoing debate in the literature (Stein, Hebart & Sterzer, 2011; Gayet, Van der Stigchel & Paffen 2014); Regarding "conscious perception", we were hesitant to discard this term, given that it is widely used in the literature, in prominent publications by leading researchers (for example, in all following papers, these terms are included in the title: "conscious perception": Block, 2014; Odegaard, Knight, & Lau, 2017; Rutiku, Aru, & Bachmann, 2016; Safavi, Kapoor, Logothetis & Panagiotaropoulos, 2014; Libedinsky & Livingstone, 2011; Railo, Koivisto, & Revonsuo, 2011; "unconscious perception": Salti, Monto, Charles, King, Parkkonen & Dehaene, 2015; Snodgrass, Bernat, & Shevrin, 2004; Merikle, 1998;). We thus think it is preferable to keep this term to better orient the readers towards relevant literature.

(B) Readily acknowledging the challenges involved in obtaining such data, it must be wondered whether the sample size is sufficient to support the conclusions. From MTL the dataset consists of just 31 units were active during rivalry or replay, but only 19 were active during both conditions. Meanwhile, 10 were active exclusively in rivalry and only 2 exclusively in replay. These are small numbers, but it is also curious that 12 of the neurons distinguished the stimulus presentation condition. This indicates that something more than "rivalry" and "replay" is being noticed. The medial frontal sample consists of 39 units. Other peculiarities can be noticed. For example, in the bottom raster of Fig 2c during rivalry, a clear change of discharge rate precedes the synchronization time by ~800 ms. Such alignment can also be observed in the top raster. How common was this? More generally, methods have been developed to quantify modulation in single spike trains. The results would be enhanced by application of such methods.

>> This comment entails different points, which we address separately:

First, we agree with the reviewer that the yield of responsive units was relatively low and now explicitly address it in the manuscript (lines 152-157). Notably, such a low yield is typical of pre-screening based studies (see Reber et al, *PNAS*, 2017; Nir et al, *Nat. Med.*, 2017; Quiroga et al, *PNAS*, 2008; Quiroga et al, *Nature*, 2005, all reporting yield of responsive units that is similar to ours), as often units that responded during the pre-screening session are not present anymore or show a different selectivity in the main experimental session that takes place a few hours later. In our case, an additional factor was the demanding nature of the task and the long duration of each rivalry and replay block, which given time limitation with the patient, allowed us to present only a few pairs of images (in some cases only one pair) in the rivalry paradigm.

Importantly, the statistical examination of the effects was not done on units but on "active traces" (lines 213-217), resulting in larger N in the analyses (36 traces for rivalry and 26 for replay). Yet to directly address this concern we conducted a Bayesian analysis on the comparison between MTL anticipatory activity onset time during rivalry and replay, to

assess whether there is sufficient data to justify a conclusive effect. This comparison yielded a Bayes factor of $BF=1123$, suggesting that the Hypothesis of earlier anticipatory activity during rivalry compared to replay is more than a thousand-fold more likely than the absence of a difference between the two conditions. This analysis was added to the manuscript (lines 218-222).

A second point raised by the reviewer is that something more than “rivalry” and “replay” might affect the responses, given that some neurons respond only to the rivalry, and others only to replay. Firstly, we can never rule out other factors at play in these experiments that we did not control for. Furthermore, our analysis was rather conservative in order to avoid false positives, but this also means that it might have missed weak responses. Most of the units that are reported to respond exclusively in rivalry or replay actually responded to some degree in both conditions but one of the responses was not strong enough to be detected by our algorithm. The figure to the right shows the average PSTH for the $n=14$ traces for which our analysis recognized responses in rivalry but not in replay, illustrating this point (conventions as in Fig. 4a). Note that Supp. Fig. S7, which shows the firing rate time courses for all rivalry and/or replay active traces in the same order also demonstrates this phenomenon. We now explicitly mention it in the legend.

Finally, the reviewer was concerned about earlier changes in firing rate that appear in the data. To address this concern, we conducted an automatic response detection analysis (Mormann et al, 2008; Hanes, Thompson and Schall, 1995) to look for changes in firing rate in the $[-3s, -1.5s]$ time window relative to the perceptual transition onset report. The results of this analysis for the example mentioned by the reviewer (Fig. 2c) are shown in the figure to the right (conventions as in Fig. 2c; detected changes in firing rate are highlighted in yellow). Overall such early changes in firing rate were detected in $24 \pm 16\%$ of rivalry trials. Such changes might represent activations

induced by the previous switch. Indeed in $25 \pm 11\%$ of rivalry trials previous dominance onset (button press) was inside this $[-3s, -1.5s]$ time window. We now explicitly address these in the manuscript (lines 618-622). Note that such early changes are randomly distributed and would not be detected by our permutation analysis that is designed to detect temporally consistent changes. Moreover this analysis was confined to the $[-1.5$ to $+1s]$ time window to minimize influences of previous behavioral events.

We also implemented the more general suggestion of the reviewer and conducted the same trial-by-trial automatic response detection analysis on the MTL data. Panel (a) of the Figure below demonstrates the results of this analysis for the unit that appears in Fig. 2 (detected responses are highlighted in yellow, median response onset time marked by purple and cyan triangles, for rivalry and replay, respectively). The results were similar to the main analysis we conducted, with earlier activity onset in rivalry ($-526 \pm 214ms$) compared to replay ($-278 \pm 300ms$; $t(23)=4.39$, $p=0.0002$, $95\% \text{ CI} = [-341, -123ms]$, two-tailed paired t-test; see panel (b) in the Figure below), which further strengthens our claims. Note however that this analysis was not optimal for our data, for two reasons. First, there is no clear baseline period (as subjects are continuously viewing the two images and keeping them in mind), which is essential for the response detection algorithm (indeed, even in the example below, which has a relatively clean baseline, some of the responses were missed). Second, this method cannot detect negative responses (i.e., decrease below baseline firing rate), which are part of the responses we found in the data, especially in frontal areas. Thus, we added this analysis as a confirmatory post-hoc analysis to our main analysis (lines 223-234), and this Figure as a Supp. Fig. S4

(C) The mechanisms mediated by the modulations are entirely unclear. The authors emphasize one preferred interpretation, but viable alternatives are not considered or not ruled out conclusively. The authors state (line 343ff) that “internal changes in the content of perception may be heralded by early activity in a cortical network that includes the ACC and preSMA”. How does “heralding” work? What is the computational theory of “heralding”? It is a poetic and evocative word, but it has no scientific content.

>> This was indeed missing and we now added a new paragraph, which discusses how the results fit with computational models of binocular rivalry (lines 417-435). We also removed the word “herald”. However, we still consider these suggestions with caution given the limited scope of the data. Notwithstanding its uniqueness, our investigation was confined by clinical considerations, we only had access to specific brain areas thus could not provide a full description of the neural events underlying binocular rivalry in the human brain. Thus, we feel that our ability to make strong claims about mechanisms is somewhat limited, and explicitly acknowledge this in the manuscript (lines 435-438).

(D) The novelty of the findings and their relation to previous observations is less clear than it could be. The authors claim their results can resolve “conflicting findings” in a “debate ... focused on the role of frontal areas in internally driven perceptual alternations”. This claim is challenged by the fact that the authors’ data is from a part of frontal lobe different from those described in previous studies. Also, the authors seem to overlook monkey single unit recordings addressing neural correlates of subjective awareness in the frontal eye field (Thompson & Schall 1999, 2000; Libedinsky & Livingstone 2011).

>> First we would like to note that the general debate about the role of prefrontal cortex in conscious perception seems not to be restricted to specific regions (e.g. *J Neurosci* recent dual perspective (2017): “Are the Neural Correlates of Consciousness in the Front or in the Back of the Cerebral Cortex? Clinical and Neuroimaging Evidence” by Boly et al vs. “Should a Few Null Findings Falsify Prefrontal Theories of Conscious Perception?” by Oddegard Knight and Lau), and previous studies have shown that patients with prefrontal lesions show abnormal transitions in bistable situations (Ricci & Blundo, 1990; Meenan & Miller, 1994; but see a case study: Valle-Inclan and Gallego, 2006) without localizing the impairment to specific regions, thus single unit results from any frontal region might be of interest. Regarding the more specific debate about the role of frontal areas in internally driven perceptual alternations, indeed as electrodes localizations are based on clinical considerations, we unfortunately did not have access to LPFC regions, such as the inferior frontal cortex (IFC), which have been implicated in the difference between rivalry and replay transitions. Yet the anterior and medial cingulate areas were also implicated in some previous fMRI studies of binocular rivalry and other bistable phenomena (e.g. Lumer, Friston & Rees, 1998; Knapen et al, 2011; Roy et al, 2017, Sato et al, 2004; Kondo & Kashino 2007). Notably, both ACC and preSMA are anatomically connected to the IFC (Aron et al, 2007), which is the area most consistently implicated in perceptual transitions (Brascamp et al, 2018), and activity in these regions has been shown to actually *precede* that of IFC in bistable and conflict monitoring situations (e.g. Swann et al, 2012; Kondo & Kashino 2007). More generally, it should be noted that imaging results are not always consistent with single-unit findings (e.g. the effects of attention on V1 responses reviewed in Boynton, 2011; V1 activity in binocular rivalry, Maier 2008), and indeed frontal areas that were found to be involved in subjective awareness in single-unit studies (e.g. FEF) do not consistently

show up in imaging studies. This is true also for other “textbook” functions of PFC that are sometimes difficult to localize using univariate and even multivariate approaches in fMRI data (see discussion by Oddegard, Knight and Lau, 2017). As ACC and preSMA have not been studied at the level of single units in this context we think our results do contribute to this ongoing debate. We now discuss this issue explicitly in the manuscript (lines 417-429).

(E) Ultimately, the theoretical insights offered by the findings are minimal. The Discussion is a meandering summary of the results and the authors’ preferred interpretation. No alternative hypotheses are seriously excluded. The take-away stated by the authors in the Abstract is simply a restatement of the observation.

>> See our reply to comment (C) above; we added a new section to strengthen the theoretical aspects of the manuscript. Specifically, about alternative hypotheses, we discuss the possibility of a cortical oscillator instigating the perceptual transitions in the manuscript, now in more detail (lines 404-416). If the reviewer would like to suggest other alternatives that we failed to consider, we are more than happy to address those in the manuscript as well.

OTHER COMMENTS

(1) Can the participants distinguish actual rivalry from replay alternations? Do they know which they are experiencing? In 2AFC condition, could they (or anyone else) distinguish the two stimulus conditions? If so, the reliability of the control condition can be questioned.

>> As opposed to most binocular rivalry studies that used an “instantaneous replay” condition (e.g., Leopold & Logothetis 1996; Tong et al, 1998; Frassle et al 2014), we acknowledged the importance of a more closely matched experience in rivalry and replay (Knapen et al, 2011) and designed a replay condition that would more closely mimic rivalry, focusing mainly on the gradual nature of the transitions and their durations. However, since the rivalry experience is distinctive and often fragmented and wave-like (Wilson, Blake and Lee, 2001), it is very difficult, if not impossible, to reach an indistinguishable replay condition (especially with the various images used in our study; for example, eyes tend to appear first when a face image is rivaling), and this problem is inherent to all binocular rivalry studies. It is important to note that patients were not informed about the difference between the rivalry and replay blocks and performed the same task. They were also not debriefed about their ability to distinguish the rivalry and the replay conditions in order to keep them naïve for potential additional sessions (we added this info to the text lines 98-100). While we believe an experienced observer could probably perform above chance in a 2AFC task, we do not think this was the case for most patients, as they were novices, and never experienced or even heard about rivalry before. The entire situation was weird and surprising for them, thus we believe that the phenomenal difference between the conditions was unlikely to be spotted. However, we now acknowledge this potential concern in the discussion (lines 370-376).

- (2) The logic of measuring the beginning of a transition is clear, but the practice can be problematic. In measuring the beginning of the transition period, how did the authors instruct or train the participants to judge this reliably? This is particularly problematic because the displays subtended 5 degree of visual angle, which supports piecemeal rivalry, especially with complex object and faces as used.

>> We instructed the patients to “release the button as soon as something in the image starts to change” (line 547 in methods; now elaborated the description of this instruction in the manuscript lines 79-85) and in our interaction with them, we realized that this was actually the easier instruction to follow. While often difficulties arose in determining the exact point in which an image became dominant (e.g., differentiating between a state in which the image is 90% dominant to 100% dominant), it seems like it was much easier for them to pinpoint the moment where a change – irrespective of its magnitude – occurred. Thus, as soon as they detected any fraction of the other image, they released the button. We appreciate the general problem of report across binocular rivalry studies; it is indeed not easy to track one’s perceptual experience and continuously report it – but we don’t think it is harder to detect the transition than the onset of the dominance; if anything, we felt it was easier for the patients. Following this comment, we now explicitly discuss it in the manuscript (lines 349-352).

- (3) On lines 338ff the authors argue that medial frontal activation was not observed in fMRI studies of bistable perception because they found both decreases and increases of discharge rate, which would average out in an fMRI voxel. This argument seemed flawed. Both increases and decreases of discharge rate entail metabolic processes utilizing oxygen, so why wouldn’t BOLD increase?

>> We decided to remove this argument from the discussion, especially given the change we already made in presenting the literature about frontal activations following a comment by reviewer 2.

- (4) The quality of unit isolation was not documented.

>> This was indeed missing in our manuscript. We now conducted a set of analyses in order to assess the quality of unit isolation and added a new methods section describing the analyses and a Supplementary Figure (Supp. Fig. S2; attached below for convenience; red lines denote the mean value for clusters identified as single-units) reporting the results. Several measures are included: proportion of inter-spike intervals which are shorter than 3ms, mean firing rates, signal to noise ratio (Joshua, Elias, Levine & Bergman, 2007), and the L-ratio (Schmitzer-Torbert & Redish, 2004) which is a measure of amount of contamination of a given cluster based on the Mahalanobis distance of spikes not included in the cluster from the center of the cluster divided by the total number of spikes in the cluster.

(5) Typographical errors are sprinkled through the manuscript.

>> We carefully re-read the entire manuscript and fixed all the errors we could find.

Reviewer #2 (Remarks to the Author):

This is an interesting manuscript that reports a technically complex experiment with a simple and potentially important finding. The experiment involved intracranial recording of single neurons in the human medial temporal lobe, anterior cingulate or pre-SMA while participants reported binocular rivalry or a matched replay condition. Participants' reporting included explicit tracking of both the start and finish of the transition period between each monocular image (or binocular replay transition). This allowed the authors to measure and analyse single unit firing time-locked to the start of the rivalry (or replay) transitions.

>> We thank the reviewer for acknowledging the importance of our findings and the challenges in obtaining them, and for the thoughtful comments that helped us improve the manuscript.

Major comments

1. Line 137 (and elsewhere) the use of ‘emergence’ is confusing, as if I have understood correctly this is the time point at which the participant judges that the dominant image is starting to transition. A second button is then subsequently pressed after this button release when the other monocular image is fully visible. ‘Emergence’ might properly be used to refer to this second timepoint at the end of the transition period. It would be helpful to have figure 1 annotated to clearly describe which time point was used for averaging of responses, and perhaps drop the use of ‘emergence’ which is ambiguous concerning whether it is the start or end of emergence.

>> We thank the reviewer for this helpful suggestion, which we followed by annotating Figure 1 (gray rectangles) and better explaining our usage of the word ‘emergence’ in the text (lines 79-85). We further substantially minimized our usage of the word, and mostly replaced it with “perceptual transition onset”.

2. One of the challenges of interpreting anticipatory activity is that the process of rivalry is continuous. For the ACC/pre-SMA activations in particular, is it possible that the premonitory activity for a button release is in fact related to the previous button press timing?

>> Indeed this was one of the biggest challenges in this study: since there is no clear baseline, it was harder to detect responses using the traditional methods. But specifically for the above concern, we do not think this is the case: if indeed the activations we observed were remnants of the button press, they should have (a) had an earlier peak with respect to the button release than with respect to the button press (since the press precedes the release); (b) been weaker/less aligned for the release as compared to the press. The data shows the opposite pattern of results as evident when comparing Fig. 5 (responses aligned to the button release) with Supp. Fig. S14 (responses aligned to the button press), relevant parts of the figures appear below for convenience.

3. It is notable that the study found differences in the timing of anticipatory activity between rivalry and replay, but also found differences in how transition periods were reported in rivalry and replay and differences in the dominance durations between rivalry and replay (despite the efforts to match these conditions). Could the differences in anticipatory activity be related to these timing differences? One way to examine this – if possible – might be to examine correlations on a trial-by-trial basis between the timing of the anticipatory activity and the (trial-by-trial) measurement of the transition period (or immediately preceding dominance period) duration.

>> Indeed despite our efforts to match the rivalry and replay conditions, there was a

significant difference in dominance durations between them, and we mention in the text that this might stem from the time it takes the change in transparency to become noticeable on both ends of the replayed transition (lines 137-143). However, we don't think it is likely that the difference in neuronal activity onset simply reflects the difference in dominance durations between the conditions, since the anticipatory activity was longer (i.e. earlier) during rivalry, where dominance durations were actually *shorter*. Yet to make sure that there is no such connection we followed the reviewer suggestion and conducted for each unit a correlation analysis between rivalry trial-by-trial perceptual transition activity onset time (as determined by the trial-by-trial response detection analysis we added), and the preceding dominance period duration. The results are presented in the figure below (one graph for each unit; dominance duration on the x-axis and activity onset on the y-axis, all trials for which the response detection algorithm recognized activity onset are denoted by blue circles, with a linear regression line and r^2 values). No significant correlation was found, mitigating the reviewer's concern and strengthening our claims. We now report this analysis in the manuscript (lines 260-268).

4. Line 337-338. Scholarship concerning the human BR studies needs improving. I had a look at reference 13 (Lumer et al) and could clearly identify ACC activity in Figure 2 and this is reported in Table 2. This is the opposite of how this reference is cited. Similarly, reference 14 Figure 3 describes activity in left SMA which is quite close to pre-SMA. I suggest this paragraph is rewritten to accurately report the references.

>> We followed the reviewer's suggestion and completely changed this paragraph to better reflect the literature (lines 402-414). Note that although Lumer et al (1998) identified ACC activity in rivalry, it did not show up in the rivalry > replay comparison, at least at the statistical threshold chosen in the paper. Same is true for left SMA in Weilenhammer et al (2013).

5. Line 379. It would be helpful to have a flow diagram to understand how selective the ultimate findings were. How many units were recorded, then how many were classified as single units, then how many were selective and proceeded to the BR sessions, then how many were actually entered into the analysis. Such a diagram/table would help the reader understand how representative the neurons that are reported might be with respect to the overall populations recorded

>> This information appears in Supp. Table 1. Following the reviewer's comment we edited the table legend to better orient the readers. We are happy to convert the table to a flow diagram if the reviewer finds it more appropriate.

6. Lines 398-410. Two different methods of binocular rivalry presentation were employed. As the separation between the two visual streams is likely to be complete with the mirror stereoscope but incomplete with the red-blue goggles (which will also attenuate the images somewhat) it would be helpful to confirm whether there is any difference in the key findings when the data are split according to visual stimulation technique.

>> We thank the reviewer for this helpful suggestion and added information about which method was used in each patient to the methods section (lines 505-506). As there are very few units from the first few patients in which we used the mirror stereoscope we cannot quantitatively compare the results under the two stimulation techniques. We switched to the red-blue goggles because the mirror stereoscope was cumbersome to use in the clinical setting, and the red-blue goggles were more convenient and familiar for the patients. Notably, the possibility of incomplete separation of the red-blue goggles would actually make it more difficult to find changes in firing locked to the perceptual transitions. We mention it clearly in the methods section now (lines 516-519).

Reviewer #3 (Remarks to the Author):

This manuscript, 'Human single neuron activity precedes the emergence of conscious perception', describes the authors efforts to provide evidence of single neuron involvement in subjective perception during binocular rivalry. The recordings were obtained from microwires implanted in neurosurgical patients who were being monitored for epilepsy. The authors were specifically interested in demonstrating changes in spiking activity in the medial temporal lobe (MTL) and in frontal lobe structures (the ACC and SMA) prior to the moment when individuals

consciously perceive a change in an image.

This builds upon previous work demonstrating changes in neural activity along the visual processing pathways during binocular rivalry. The authors extend this work in several important ways. First, they demonstrate that changes in MTL neuronal activity precedes perception. The MTL in many ways sits at the top of the visual hierarchy, and so this work suggests. Second, they provide a novel control condition that is matched for the duration of transitions, which they call replay, and show that the timing of neural responses are slower in this replay condition. This suggest that internally driven transitions involve earlier neuronal activity. And third, they provide novel evidence that neurons within the ACC also exhibit early responses to internally generated transitions. In this sense, then, this work collectively begins to address the question of how humans create a subjective conscious experience.

The manuscript is well written and clear, and I believe the results are compelling. The statistical methods used to demonstrate changes in firing rates from baseline are appropriate and well done. I believe that the central interpretation, that MTL neuronal activity precedes subjective transitions is well supported. I also believe that the same claim about ACC neurons is also supported. I have the following concerns that, if addressed, could improve the overall manuscript:

>> We thank the reviewer for clearly pointing the strengths and significance of our study, and for the helpful suggestions that indeed strengthened the manuscript.

Major Concerns

Single unit recordings from the human brain are quite challenging, and one of the issues to consider is the fact that a given unit recorded on an electrode during one experimental session may not be present during a separate session recorded even just hours later. In their experimental setup, each test of BR is preceded by a separate experimental session to identify unit responses to different images. Isn't there a concern that the units identified during these initial localization sessions would be different than the ones used in the BR experiments? How do the authors account for this?

>> Indeed, the pre-screening session that was used to select the images for the binocular rivalry session took place a few hours earlier. To directly address the possible concern raised by the reviewer, the rivalry session itself also included a non-rivalrous condition. Each rivalry block was preceded by a non-rivalrous presentation of the images, so we could be sure that we are looking at the same neurons across the rivalrous, non-rivalrous and replay conditions. We now clarified this point in the manuscript and methods (lines 74-76; 495-497). As the reviewer suggests, some of the selective responses we identified in the pre-screening session were indeed not there during the actual rivalry session. In such cases, we had no responses in the non-rivalrous condition and these neurons were not further analyzed – this is part of what made the yield of the neurons relatively low, and the data collection very challenging.

In identifying units that are responsive to individual images, and therefore to determine which image is the preferred image for each pair, it appears that the authors treat both increasing and decreasing responses in the same way. This is relevant when looking at both the individual and

population histograms, since the authors then invert any negative changes in firing rates to determine population level responses (Fig 4). However, it is unclear how to interpret this and how often this occurs. I think that that one interpretation is that there are changes in neural activity that precede transitions regardless of direction, but it would be important to clarify and demonstrate how many of these responses are actually decreases in spiking activity. Also, it would be helpful if the authors could comment on how such selective decreases in activity might play a role in such conscious perception.

>> We thank the reviewer for this helpful suggestion. Selective decreasing firing rate responses are often found in human MTL neurons (e.g. Fried et al., *Cerebral Cortex* 2002). To better demonstrate this type of responses we added a supplementary figure depicting an amygdala unit that responds selectively to one of the images by decreasing its firing rate (Supp Fig. S3), and referred to it in the text (lines 182-185). Additionally we now mention in the text the number of such responses (lines 155-158; 289; 306; Fig. 4 legend). Note that we inverted the responses only for the population average, while the individual units' histograms present the data as-is. The units that were inverted are easy to spot (highlighted part is blue) on the color traces parts of Fig. 4 (panels b & c) and Supp. Figs. S6 & S7. We also added a sentence to the legend that explicitly explains this point.

Regarding our interpretation of this firing pattern, we think of an MTL representation as a neuronal network in which some of the participating neurons selectively increase their firing rate while others decrease it. Appearance of typical activity (either FR increase or decrease) in any part of this network might reinstate the activity in other parts of the network and lead to the emergence of the represented "concept". The important point is that the same type of response was typically found in normal viewing (non-rivalrous condition) and before the transition in the rivalry condition. Similarly, frontal responses – whether positive or negative, can represent an attentional signal that biases the competition in MTL or lower-level visual areas. We now discuss these interpretations explicitly in the manuscript (lines 425-433).

There is the possibility that an image is dominant, and then loses dominance only to immediately regain it. I think most of the text is written to consider only the possibility that the dominant image switches. But what happens in these cases when dominance is lost, only to be regained by the same image immediately afterwards? This would be helpful to know, as the question would be whether the same neurons become active once again in this situation. For example, suppose a non-preferred image is dominant, and suppose that a button release is initiated, followed by the same button press later (indicating that the same non-preferred image is dominant again). During that transition, are there still changes in spiking activity related to the preferred image? Perhaps. However, in the converse situation, when the preferred image is dominant, then transitions, and then is dominant again, I might expect not to see changes in spiking activity associated with that non-preferred image during the transition.

>> Indeed such cases of incomplete switching occurred in 24 % of the trials. Note however that our analysis focuses on the beginning of the switch: the emergence of the non-dominant image, even transiently, should be accompanied (or result from) a reinstatement of the neuronal representation of that image. Therefore, in our analysis we look at complete and incomplete transitions together (i.e. all the times when the subject released the "other button" to indicate that the relevant image starts to emerge. It is true that this activity

might be shorter or weaker if the image doesn't gain full dominance, but it should still be there, and therefore it is included in the analysis. Note that neural activity peaks well before dominance onset (Supp. Fig. S13) and returns to baseline shortly after it (in accordance with normal viewing responses which usually ends after 600-800ms after image onset, while the image is still on screen, see non-rivalrous responses), therefore we expect the activity to diverge from baseline again when the dominant image regains dominance even after losing it momentarily. We now give information about this type of reversals in the manuscript and explain our decision to include them in the analysis (lines 119-121; 162-166; 596-598).

The statistical testing comparing rivalry to replay activity at the level of the population responses is sound, and the bootstrapping analysis provides some validation to these results. However, it would be important to know whether this effect is driven by only a small subset of neurons within only one or two individuals, or whether it is more evenly distributed. Table 1 would suggest that most of the data are captured from only a small number of individuals. As such, I would suggest that the authors perform these statistical comparisons both at the level of individual units (they could use a permutation test comparing rivalry to replay conditions), and for units aggregated within individual subjects, and report how frequently this effect arises.

>> Indeed we conducted the bootstrapping analysis in order to address this point. Following the reviewer's comment, we subjected the results of the new trial-by-trial response detection analysis to a Linear Mixed Model analysis with condition (rivalry or replay) as a fixed factor and patient and trace as random factors with separate slopes and intercepts, and compared it to a model with the random factors only. The full model better explained the data: $\chi^2(1)=3.75$, $p=0.053$. Models with either patient *or* trace as exclusive random factors gave similar results, and similarly suggested that the difference between rivalry and replay onsets does not stem from specific patients or traces: $\chi^2(1)=3.86$, $p=0.049$ and $\chi^2(1)=14.03$, $p=0.0002$ for patient and trace, respectively.

The results describing neuronal activity in the ACC are also compelling, and I think the interpretation that neuronal activity in this region precedes these transitions is also well supported. However, I do think that one of the central claims of the manuscript, that ACC activity precedes MTL activity is hard to justify, primarily because there is no direct comparison within an individual patient between ACC and MTL activity. As such, there is no direct statistical test between them. I think the authors can still support most of the claims in the manuscript, but I think they should consider tempering the specific claim that ACC activity precedes MTL activity.

>> Following the reviewer's comment we conducted an additional analysis on MTL units only from the three patients that had frontal electrodes. The results (presented in the figure below) show the same difference between MTL and ACC/preSMA as when all units were included, suggesting that this difference does not stem from differential response times between patients. This is now added as a new supplementary figure (Supp. Fig. S12).

Are there any differences in MTL neuronal activity between the different subregions?

>> Following this comment we repeated the analysis for each of the MTL subregions separately and added a supplementary figure (Supp. Fig. S5) with the average response profiles (also attached below). However, because the number of units per region is small and since most patients didn't have responsive neurons in more than one or two regions, we can't reliably quantify the difference between different subregions. We address it in the text now (Supp. Fig. S5 legend).

The authors raise the issue of ocular dominance, and state that the subjects' percepts were equally distributed between the two images. However, while the centers of the gamma distributions used to illustrate the frequency of relative image dominance lie around 1, the distributions clearly have longer tails to the right. This is clear in the distribution of transitions shown in Fig 1c, where most of the percepts are for the image of the snakes. I don't think this affects the main conclusions of the paper, but it would worth discussing the issue of ocular dominance in the context of these results.

>> Note that images were switched between the eyes in the middle of each rivalry block to avoid unbalanced durations due to ocular dominance. This appeared in the methods but now we also state it clearly in the main text (lines 125-127). The average predominance score was actually remarkably close to 50% (predominance score= 0.50 ± 0.05 ; $t(53)=0.40$; $p=0.69$; 95% CI = [0.45 0.53]; two-tailed t-test against 0.5; line 124). Though dominance durations were not always perfectly equally distributed, they were fairly close (in the example given by the reviewer, the dominance score of the snakes images was 52%), and in any case we do not directly compare neural activity for the two images of a given pair, but denote one as the "unit's preferred image" and look at the activity preceding its emergence. Thus we don't think unbalanced duration distribution, when exists, could affect the results.

REVIEWERS' COMMENTS:

Reviewer #1 (Remarks to the Author):

The authors have responded to reviewer comments thoroughly and effectively. Thank you.

Acknowledging the limitations of the data and the qualifications of the conclusions, enthusiasm for this current version is high.

Reviewer #2 (Remarks to the Author):

I thank the authors for considering and responding to my comments. I have reviewed their responses and the revised manuscript which is considerably improved by an attentive revision. I have no further concerns to raise.

Reviewer #3 (Remarks to the Author):

I commend the authors on an extensive revision of their manuscript that has addressed all of my primary concerns and that has improved the quality of the manuscript. In addition, it appears that the authors have also tempered some of their claims for which the data were not fully conclusive. I think this is a good study with good statistical support, and I would support publication based on this revision.